

**Efficient N₂O₅ Uptake and NO₃ Oxidation in the Outflow of Urban Beijing**
Haichao Wang[1], Keding Lu[1*], Song Guo[1], Zhijun Wu[1], Dongjie Shang[1], Zhaofeng Tan[1], Yujue Wang[1],
Michael Le Breton[2], Wenfei Zhu[3], Shengrong Lou[3], Mingjin Tang[4], Yusheng Wu[1], Jing Zheng[1], Limin
Zeng[1], Mattias Hallquist[2], Min Hu[1] and Yuanhang Zhang[1, 5]
[1]State Key Joint Laboratory or Environmental Simulation and Pollution Control, College of
Environmental Sciences and Engineering, Peking University, Beijing, China.
[2]Department of Chemistry and Molecular Biology, University of Gothenburg, Gothenburg, Sweden
[3]Shanghai Academy of Environmental Sciences, Shanghai, China
[4]State Key Laboratory of Organic Geochemistry and Guangdong Key Laboratory of Environmental
Protection and Resources Utilization, Guangzhou Institute of Geochemistry, Chinese Academy of
Sciences, Guangzhou, China
[5]CAS Center for Excellence in Regional Atmospheric Environment, Chinese Academy of Sciences,
Xiamen, China
*Corresponding to: Keding Lu (k.lu@pku.edu.cn)
**Abstract.** Nocturnal reactive nitrogen compounds are important for understanding regional air
pollution. Here we present the measurements of dinitrogen pentoxide (N₂O₅) associated with nitryl
chloride (ClNO₂) and particulate nitrate (pNO₃⁻) in a suburban site of Beijing in the summer of 2016.
High levels of N₂O₅ and ClNO₂ were observed in the outflow of the urban Beijing air masses, with 1-
min average maxima of 937 pptv and 2.9 ppbv, respectively. The N₂O₅ uptake coefficients, γ, and
ClNO₂ yield, f, were experimentally determined from the observed parameters. The N₂O₅ uptake
coefficient ranged from 0.012 to 0.055, with an average of 0.034 ± 0.018, which is in the upper range
of previous field studies reported in North America and Europe but is a moderate value in the North
China Plain (NCP), which reflects efficient N₂O₅ heterogeneous processes in Beijing. The ClNO₂ yield
exhibited high variability, with a range of 0.50 to unity and an average of 0.73 ± 0.25. The nighttime
nitrate radical (NO₃) was calculated assuming that the thermal equilibrium between NO₃ and N₂O₅
was maintained. In NOₓ-rich air masses, the oxidation of nocturnal biogenic volatile organic
compounds (BVOCs) was dominated by NO₃ rather than O₃. The production rate of organic nitrates
(ONs) via NO₃+BVOCs was significant, with an average of 0.11 ± 0.09 ppbv h⁻¹. We highlight the
importance of NO₃ oxidation of VOCs in the formation of ONs and subsequent secondary organic
aerosols in summer in Beijing. The capacities of BVOCs oxidation and ONs formation are maximized
and independent of NOx under a high NOₓ/BVOCs ratio condition (>10), which indicates that the
initial reduction of the NOx emission cannot help reduce the nocturnal formation of ONs.





**1. Introduction**
It has been well recognized that reactive nitrogen compounds, specifically the nitrate radical ($NO_3$)
and dinitrogen pentoxide ($N_2O_5$), play a key role in nighttime chemistry (Wayne et al., 1991; Brown
and Stutz, 2012). $NO_3$ is the most important oxidant in the nighttime and can be considered the
nighttime analogue of the hydroxyl radical (OH) for certain VOCs (Wayne et al., 1991; Benton et al.,
2010). $NO_3$ can initiate the removal processing of many kind of anthropogenic and biogenic emissions
after sunset. In the $NO_X$-rich plumes, $NO_3$ is responsible for the vast majority of the oxidation of
biogenic VOCs because of its rapid reactions with unsaturated hydrocarbons (Edwards et al., 2017).
$NO_3$ is predominantly formed by the reaction of $NO_2$ with $O_3$ (R1) and further reacts with $NO_2$ to
produce $N_2O_5$ (R2). Because $N_2O_5$ is rapidly decomposed back into $NO_2$ and $NO_3$ (R3), $NO_3$ and $N_2O_5$
are in dynamic equilibrium in the troposphere.
$NO_2 + O_3 \rightarrow NO_3 + O_2$ (R1)
$NO_2 + NO_3 + M \rightarrow N_2O_5 + M$ (R2)
$N_2O_5 + M \rightarrow NO_2 + NO_3 + M$ (R3)
Photolysis of $NO_3$ and its reaction with NO are rapid, which leads to a daytime $NO_3$ lifetime being
shorter than 5 s with extremely low concentrations, whereas in low-NO air masses, the fate of $NO_3$ is
mainly controlled by the mixing ratios of various VOCs and $N_2O_5$ heterogeneous hydrolysis because
the two terms are the dominating loss pathways of $NO_3$ and $N_2O_5$. The VOCs reaction is significant
downwind of a human-dominated area or a strongly urban-influenced forested area in summer. The
$NO_3$ oxidation of VOCs was responsible for more than 70% nocturnal $NO_3$ loss in Houston (Stutz et
al., 2010) and contributed approximately 50% in the forest region in Germany (Geyer et al., 2001).
The reactions of $NO_3$ with several BVOCs produce considerable organic nitrates (ONs) with efficient
yields, which act as important precursors of secondary organic aerosols (SOA). The reaction of $NO_3$
with isoprene has a considerable SOA yield of 23.8% (Ng et al., 2008), and the reaction with
monoterpene, such as limonene, can reach 174% at ambient temperatures (Boyd et al., 2017). The
reactions of $NO_3$+BVOCs are critical to the studies of aerosols on regional and global scales (Fry et
al., 2009; Rollins et al., 2009; Pye et al., 2010; Ng et al., 2017). For example, ONs had extensive
percentages of fine particulate nitrate ($pNO_3^-$) (34% - 44%) in Europe (Kiendler-Scharr et al., 2016).
The heterogeneous hydrolysis of $N_2O_5$ produces soluble nitrate ($HNO_3$ or $NO_3^-$) and nitryl chloride
($ClNO_2$) on the chloride-containing aerosols (R4) (Finlayson-Pitts et al., 1989). This reaction is known
to be an important intermediate in the $NO_X$ removal processes (Brown et al., 2006). The rate coefficient
of the heterogeneous $N_2O_5$ reaction is given in Eq. 1 (Tang et al., 2017).
$N_2O_5 + (H_2O \text{ or } Cl^-) \rightarrow (2 - f) NO_3^- + f\, ClNO_2$ (R4)
$k_{N2O5} = 0.25 \cdot c \cdot \gamma(N_2O_5) \cdot S_a$ (Eq. 1)
where $c$ is the mean molecule speed of $N_2O_5$, $S_a$ is the aerosol surface concentration and $\gamma(N_2O_5)$ is the
$N_2O_5$ uptake coefficient. $N_2O_5$ heterogeneous hydrolysis is one of the major uncertainties of the $NO_3$
budget since the $N_2O_5$ uptake coefficient can be highly variable and difficult to quantify (Brown and
Stutz, 2012; Chang et al., 2011; H. C. Wang et al., 2016). Laboratory and field measurement studies
have reported that the $N_2O_5$ uptake coefficient has large variability and ranges from <0.001 to 0.1; the



N₂O₅ uptake coefficient is subject to relative humidity (RH), particle morphology, compositions (water
content, nitrate, sulfate, organic or mineral particles) and other factors (e.g., Wahner et al., 1998;
Mentel et al., 1999; Hallquist et al., 2003; Thornton et al., 2003; Thornton et al., 2005; Brown et al.,
2006; Bertram and Thornton, 2009; Tang et al., 2012, 2014; Gaston et al., 2014; Grzinic et al., 2015).
The coupled chemical mechanisms in ambient conditions are still not well understood. ClNO₂ forms
and accumulates with a negligible sink during the night and further photolyzes and liberates the
chlorine radical (Cl) and NO₂ after sunset. Hundreds of pptv to ppbv of ClNO₂ can lead to several
ppbv of O₃ enhancement and significant primary ROx production (Osthoff et al., 2008; Thornton et al.,
2010; McLaren et al., 2010; Riedel et al., 2014; Sarwar et al., 2014; Tham et al., 2016).
Large amounts of NOx have been emitted for the past several decades in China, but comprehensive
field studies of the nighttime chemical processes of reactive nitrogen oxides remain sparse. Previous
studies have found high mixing ratios of NO₃ associated with high NO₃ reactivity in the megacities in
China, including Shanghai, the Pearl River Delta (PRD) and Beijing (Li et al., 2012; Wang et al., 2013;
Wang et al., 2015). N₂O₅ concentration was elevated in Beijing (H. C. Wang et al., 2017a; H. C. Wang
et al., 2017c) but was moderate in other places of North China Plain (NCP), such as Wangdu, Jinan
and Mount Tai (Tham et al., 2016; X. F. Wang et al., 2017; Z. Wang et al., 2017). Recently, the N₂O₅
uptake coefficients were determined to be very high, even up to 0.1 in NCP, but the reason is still not
well studied (H. C. Wang et al., 2017c; X. F. Wang et al., 2017; Z. Wang et al., 2017). Reactive N₂O₅
chemistry was also reported in Hong Kong, which had the highest N₂O₅ concentration (T. Wang et al.,
2016; Brown et al., 2016). Observations and model simulations revealed that fast heterogeneous uptake
of N₂O₅ is an important pathway of pNO₃⁻ formation in China (H. C. Wang et al., 2017b; H. C. Wang
et al., 2017c; Z. Wang et al., 2017; Su et al., 2017); the reaction also considerably contributed to NOx
removal (Z. Wang et al., 2017; Brown et al., 2016). Moreover, chlorine activation from N₂O₅ uptake
had a significant effect on daytime photolysis chemistry in China (Xue et al., 2015; Li et al., 2016;
Tham et al., 2016; T. Wang et al., 2016).
In this study, to quantify the contribution of NO₃ and N₂O₅ chemistry to the atmospheric oxidation
capacity and the NOx removal process in the outflow of urban Beijing, we reported the measurement
of N₂O₅, ClNO₂, and related species in the surface layer of a suburban site in Beijing and determined
the N₂O₅ heterogeneous uptake coefficients and ClNO₂ yields. The nighttime NO₃ oxidation to the
biogenic VOCs and its impact on the ONs formation in the NOx-rich region were diagnosed. Finally,
the nighttime NOx removal via the NO₃ and N₂O₅ chemistry was estimated and discussed.

**2. Method**
**2.1 The site**
Within the framework of a Sino-Sweden joint research project, "Photochemical Smog in China", a
summer field campaign was conducted in Beijing to enhance our understanding of the secondary
chemistry via photochemical smog and the heterogeneous reactions (Hallquist et al., 2016). The data
presented here were collected at a regional site, PKU-CP (Peking University Changping campus), from
23 May to 5 June 2016. The measurement site is located in the northern rural area of Beijing,




approximately 45 km from the city center; the closest road is approximately 1 km to the south, and
there are no major industry surroundings (Figure. 1). The site is surrounded to the north, east and west
by mountains. The general feature of this site is that it captures air masses with strong influences from
both urban and biogenic emissions. Instruments were set up on the fifth floor of the main building of
the campus with inlets approximately 12 m above the ground. Time is given in this paper as CNST
(Chinese National Standard Time = UTC+8 h). During the campaign, sunrise was at 05:00 CNST and
sunset was at 19:30 CNST.
**2.2 Instrument setup**
A comprehensive suite of trace gas compounds and aerosol properties was measured in the field study,
and the details are listed in Table 1. $N_2O_5$ was measured by a newly developed cavity enhanced
absorption spectrometer (CEAS; H. C. Wang et al., 2017a). In the CEAS, ambient $N_2O_5$ was thermally
decomposed to $NO_3$ in a perfluoroalkoxy alkanes (PFA) tube (length: 35 cm, I.D.: 4.35 mm) heated to
120 °C and was then detected within a PFA resonator cavity; the cavity was heated to 80 °C to prevent
$NO_3$ reacting back to $N_2O_5$. Ambient gas was sampled with a 1.5-m sampling line (I.D.: 4.35 mm) with
a flow rate of 2.0 L min$^{-1}$. NO was injected for 20 seconds to destroy $NO_3$ from $N_2O_5$ thermal
decomposition in a 5-minute cycle, and the corresponding measurements were then used as reference
spectra ($I_0$). A Teflon polytetrafluoroethylene (PTFE) filter was used in the front of the sampling
module to remove ambient aerosol particles. The filter was replaced with a fresh one every hour to
avoid the decrease of $N_2O_5$ transmission efficiency due to aerosol accumulation on the filter. The limit
of detection (LOD) was 2.7 pptv (1$\sigma$), and the measurement uncertainty was 19%.
$ClNO_2$ and $N_2O_5$ were also detected using a Time of Flight Chemical Ionization Mass Spectrometer
(ToF-CIMS) with the Filter Inlet for Gas and AEROsols (FIGAERO; Lopez-Hilfiker et al., 2014;
Bannan et al., 2015). Briefly, the gas phase species were measured via a 2-m-long, 6-mm-outer-
diameter PFA inlet while the particles were simultaneously collected on a Teflon filter via a separate
2-m-long, 10-mm-outer-diameter copper tubing inlet; both had flow rates of 2 L min$^{-1}$. The gas phase
was measured for 25 minutes at 1 Hz, and the FIGAERO instrument was then switched to place the
filter in front of the ion molecule region; it was then heated incrementally to 200 °C to desorb all the
mass from the filter to be measured in the gas phase, which resulted in high-resolution thermo grams.
Formic acid calibrations were performed daily using a permeation source maintained at 40 °C. Post-
campaign laboratory calibrations of $N_2O_5$ were first normalized to the campaign formic acid
calibrations to account for any change in sensitivity (Le Breton et al., 2014). Then, $ClNO_2$
measurements were quantified by passing the $N_2O_5$ over a wetted NaCl bed to produce $ClNO_2$. The
decrease in $N_2O_5$ from the reaction with NaCl was assumed to be equal to the concentration of $ClNO_2$
produced (i.e., 100% yield). The sensitivities of the CIMS to $N_2O_5$ and $ClNO_2$ were found to be 9.5
and 1.2 ion counts per pptv Hz$^{-1}$, respectively, with errors of 23% and 25% for $ClNO_2$ and $N_2O_5$,
respectively. The limit of detection (LOD) for $ClNO_2$ and $N_2O_5$ were 16 and 8 pptv, respectively. An
intercomparison of $N_2O_5$ measurements between the CEAS and FIGAERO-ToF-CIMS showed good
agreement; a companion paper on chlorine photochemical activation during this campaign gives
detailed intercomparison results of $N_2O_5$ measured by the two different techniques (Le Breton et al.,
154 2018).





Sub-micron aerosol compositions (PM$_{1.0}$), including nitrate, sulfate, chloride, ammonium and
organic compounds, were measured by a High Resolution Time of Flight Aerosol Mass Spectrometer
(HR-ToF-AMS) (DeCarlo et al, 2006, Zheng et al., 2017). Particle number and size distribution (PNSD)
was measured by a scanning mobility particle sizer (SMPS, TSI 3936) and an aerosol particle sizer
(APS, TSI 3321) (Yue et al., 2009). SMPS measured the particles in the range between 3.5 nm and
523.3 nm in diameter, and APS measured the particles with a diameter range from 597.6 nm to 10.0
µm. $S_a$ was calculated based on the dry-state particle number and geometric diameter in each size bin
(3.5 nm - 2.5 µm). Dry-state $S_a$ was corrected to wet particle-state $S_a$ for particle hygroscopicity by a
growth factor. The growth factor, f(RH)=$1 + 8.77 \times (RH/100)^{9.74}$, was derived from the aerosol optical
property in autumn in Beijing and is valid for 30% < RH < 90% (Liu et al., 2013). The uncertainty of
the wet aerosol surface areas was estimated to be ~30%, associated from the error from dry PNSD
measurement (~20%) and the growth factor (~20%). During this measurement, fine particles below
500 nm contributed to more than 90% of the total particle aerosol surface area.
VOCs were measured by Proton Transfer Reaction Mass Spectrometry (PTR-MS) with a time
resolution of 5 minutes (de Gouw and Warneke et al., 2007; Wang et al., 2014). A commercial
instrument (Thermo Electron model 42i) equipped with a molybdenum-catalytic converter was used
to monitor NO$_X$. The LOD were 60 pptv (1 min) for NO and 300 pptv (1 min) for NO$_2$, with both at a
20% precision (Tan et al., 2017). The molybdenum-catalytic technique not only converts NO$_2$ to NO
but also converts ambient NO$_y$ such as peroxyacetyl nitrate (PAN) and HNO$_3$. Therefore, the measured
NO$_2$ concentration corresponded to NO$_2$ + NO$_y$ and was normally higher than the real concentration,
especially in an aged air mass with high NO$_x$ conditions. In this study, we used a factor of 0.6 to correct
the nighttime NO$_2$ concentration (a detailed explanation is in the Support Information Text S1 and
Figure S1). O$_3$ was measured by a commercial instrument using ultraviolet (UV) absorption (Thermo
Electron model 49i); the LOD was 0.5 ppbv, with an uncertainty of 5%. The mass concentration of
PM$_{2.5}$ was measured using a standard Tapered Element Oscillating Microbalance (TEOM, 1400A
analyzer). Meteorological parameters included relative humidity, temperature, pressure, wind speed,
and wind direction and were available during the campaign. Photolysis frequencies were calculated
from the spectral actinic photon flux density measured by a spectroradiometer (Bohn et al., 2008).

**3. Results**
**3.1 Overview**
During the campaign, the meteorological conditions of the site were characterized by high temperature
and low relative humidity (RH); the temperature ranged from 10 - 34 °C and was 23 ± 5 °C on average,
and RH ranged from 10% - 80%, with an average of 37% ± 15%. Because of the special terrain of the
observation site, the local wind was measured by the in situ meteorological stations; the site has a
typical mountain-valley breeze that cannot reflect the general air mass movement patterns at slightly
higher altitudes. Figure 2 shows the calculated backward trajectories using the Hybrid Single-Particle
Lagrangian Integrated Trajectory (HYSPLIT) model (Draxler and Rolph, 2003); these images show
the 24-h backward particle dispersion trajectories for 12:00 local time (CNST) as the starting time
during May 23 - July 5, 2016. According to the results of HYSPLIT, the arrivals of air masses were





mainly from the northwest and the south. Therefore, we meteorologically separated the measurement
period into two parts. The first three days show that the air masses came from the north or northwest;
the air masses represent the background region (defined as Background Air Mass, BAM). The air
masses after May 26 originated from the polluted NCP and passed over urban Beijing; they were
characterized by large $NO_X$ emissions and severe photochemical pollution (defined as Urban Air Mass,
UAM).
The time series of $N_2O_5$, $ClNO_2$ and other relevant species are shown in Figure 3, and nighttime
statistical results are listed in Table S1. The daily 8-h maximum of $O_3$ concentration exceeded 93 ppbv
(Chinese national air quality standard) for 8 of 12 days, and all the $O_3$-polluted air masses came from
the urban region. When the air masses were from the background region, the daily maximum of $O_3$
was only approximately 60 ppbv, much lower than that from the urban region. The $NO_2$ concentration
was elevated, with a nocturnal average value over 10 ppbv during the urban air mass period. The
nocturnal nitrate radical production rate, $P(NO_3)$, was profound, with an average of $1.2 \pm 0.9$ ppbv h$^{-1}$
, which is comparable with rates previously reported in the NCP and Hong Kong (Tham et al., 2016;
Brown et al., 2016; Z. Wang et al., 2017; X. F. Wang et al., 2017). The daily peaks of $N_2O_5$ were 100-
500 pptv most nights; the maximum of 937 pptv in a 1-min average was observed near 20:00 on the
early night of June 2, when the $P(NO_3)$ was up to 4 ppbv h$^{-1}$. The average mixing ratio of $N_2O_5$ was 73
$\pm 90$ pptv, which is much higher than recent measurements reported in North China (Tham et al., 2016;
X. F. Wang et al., 2017; Z. Wang et al., 2017) but much lower than that observed in the residual layer
of the outflow from the PRD region, where the $N_2O_5$ was up to 7.7 ppbv (T. Wang et al., 2016). With
an elevated $O_3$ mixing ratio in the first half of the night, the NO lifetime was only several minutes, and
the mixing ratio of NO concentration was observed below the detection limit. During the second half
of the night when the $O_3$ concentration was consumed to low concentration, high levels of NO could
occasionally be observed, and $N_2O_5$ dropped to zero because of the fast titration by NO, such as the
events that occurred on the second half of the nights of May 24, 28, 30. The $PM_{2.5}$ mass concentration
was moderate during the measurement period, with an average of $26 \pm 21$ μg m$^{-3}$, and the average
aerosol surface area was $560 \pm 340$ μm$^2$ cm$^{-3}$. Elevated $ClNO_2$ was observed to have a daily maximum
1-min average of over 800 pptv during the urban air masses period; the campaign maximum of up to
2.9 ppbv was observed on the morning (05:30) of May 31, which implied that fast $N_2O_5$ heterogeneous
hydrolysis and effective $ClNO_2$ yields are common in Beijing. The level of $ClNO_2$ was comparable
with the results in NCP (Tham et al., 2016; X. Wang et al., 2017; Z. Wang et al., 2017) but slightly
higher than that measured in coastal (e.g., Osthoff et al., 2008) and inland sites (e.g., Thornton et al.,
2010) in other regions of the world.
**3. 2 Mean diurnal profiles**
The mean diurnal profiles of the measured $NO_2$, $O_3$, $N_2O_5$, $ClNO_2$ and the particle chloride content are
shown in Figure 4, as well as the calculated $NO_3$ based on the thermal equilibrium of $NO_2$, $NO_3$ and
$N_2O_5$. The left panels show the average results of the BAM period, and the right panels show those of
the UAM period. The $NO_2$ and $O_3$ from the UAM were much higher than were those from the BAM,
as were the mixing ratios of $N_2O_5$, $NO_3$ and $ClNO_2$. The daily variation tendencies of those species in
the two kinds of air masses were similar. $N_2O_5$ began to accumulate in the late afternoon and increased
sharply after sunset. The single peak occurred near 20:00 and then gradually decreased to the LOD



before sunrise; the $N_2O_5$ maxima occurred at a similar time to our previous observation in urban
Beijing (H. C. Wang et al., 2017c); however, the $N_2O_5$ decrease rate after the peak time was much
slower than that in urban Beijing, where the $N_2O_5$ dropped to almost zero in 2-4 hours, which suggests
a relatively slow $N_2O_5$ loss rate in suburban Beijing. The peaks of $N_2O_5$ during the BAM period and
the UAM period were approximately 75 pptv and 150 pptv, respectively. The calculated $NO_3$ diurnal
profile was quite similar to that of $N_2O_5$, and the peaks of $NO_3$ during the BAM and UAM periods
were approximately 11 pptv and 27 pptv, respectively.
$ClNO_2$ accumulated corresponding to $N_2O_5$ after sunset but $ClNO_2$ peaked in the middle or the
second half of the night since the nocturnal sinks of $ClNO_2$ were negligible to our knowledge. The
diurnal peak of $ClNO_2$ in the BAM period was approximately 125 pptv, whereas the diurnal peak of
$ClNO_2$ was over 780 pptv in the UAM period and approximately 6 times as high as that in the UAM
period. Particle chloride ($Cl^-$) is regarded as a key factor that affected the $ClNO_2$ yield on the aerosol
surface. Higher particle chloride led to higher $ClNO_2$ yield and promoted the $N_2O_5$ conversion to
$ClNO_2$ (e.g., Roberts et al., 2009), whereas the particle chloride content during the measurement was
below 60 pptv and was extremely lower than the mixing ratio of $ClNO_2$. The HYSPLIT model results
showed that the air masses had almost always continental conditions; as was mentioned above, fine
particles dominated the $S_a$, which meant that large amounts of the particle chloride were not
replenished by NaCl from marine sources but possibly by the gas-phase HCl (Ye et al., 2016). $Cl^-$ was
found to be correlated strongly with CO and $SO_2$, likely to originate from an anthropogenic source,
such as power plants or combustion sources (Le Breton et al., 2018). Up to 10 ppbv of HCl was
observed by a Gas and Aerosol Collector combined with Ion Chromatography (GAC-IC; Dong et al.,
2012) in the urban Beijing in September, 2016, which implies that the potential particle Cl source was
sufficient and the gas-phase HCl was possibly the main particle chloride source by the acid
replacement reaction. After sunrise, $ClNO_2$ was photolyzed and decreased with the increasing
photolysis intensity; however, the photolysis with profound $ClNO_2$ was still maintained until noon.
Similar to the studies reported in London, Texas and Wangdu (Bannan et al., 2015; Faxon et al., 2015;
Tham et al., 2016), we observed sustained elevated $ClNO_2$ events after sunrise in 5 of 12 days. For
example, on the morning of May 30, $ClNO_2$ increased after sunrise and peaked at approximately 8:00
am, with a concentration over 500 pptv, which was impossible from the local chemical formation since
$N_2O_5$ dropped to almost zero and the needed $N_2O_5$ uptake coefficients were unrealistically high.
Previous work has suggested that abundant $ClNO_2$ produced in the residual layer at night and
downward transportation in the morning may help to explain this phenomenon (Tham et al., 2016).
**3.3 Variation of $N_2O_5$ in the background air masses**
During the BAM period, the $O_3$ concentration was excessive compared with $NO_2$. In the $NO_3$ and $N_2O_5$
formation processes, the limited $NO_2$ in high $O_3$ region indicates that the variation of $NO_2$ is more
essential to the variation of the $N_2O_5$ concentration. As shown in Figure 5, during the night of May 24
(20:00 - 04:00), the local emission of NO was negligible. $O_3$ concentration was larger than 25 ppbv,
much higher than $NO_2$ and free of the local NO emission. The variation of $N_2O_5$ concentration was
highly correlated with the mixing ratio of $NO_2$ ($R^2 = 0.81$). The result suggests that when the air mass
with high $O_3$ was sampled from the background air mass, the $N_2O_5$ concentration was especially
subjected to the $NO_2$ concentration rather than $O_3$. Furthermore, The variation of $N_2O_5$ concentration





was considerably correlated with the $NO_3$ production rate ($R^2 = 0.60$), suggests the mixing ratio of
$N_2O_5$ was subject to the formation processes in clean air masses.

**3.4 Elevated $ClNO_2$ to $N_2O_5$ ratio**

Large day-to-day variabilities of $N_2O_5$ and $ClNO_2$ were observed during the measurement period.
Following the work of Osthoff et al. (2008), Mielke et al. (2013), Phillips et al. (2012) and Bannan et
al. (2015), we used the relative production rates, $ClNO_2$:$N_2O_5$, to describe the conversion capacity of
$N_2O_5$ to $ClNO_2$. The nighttime peak values and mean values were used to calculate the ratios, and two
kinds of daily ratios are listed in Table S2. The average nighttime ratio ranged from 0.7 to 42.0, with
a mean of 7.7 and a median of 6.0. The $ClNO_2$ formation was effective, with $ClNO_2$:$N_2O_5$ ratios larger
than 1:1 throughout the campaign, except for the night of May 26, when the ratio was 0.7:1. Previous
observations of the $ClNO_2$:$N_2O_5$ ratios are summarized in Table 2. Compared with the results
conducted in similar continental regions in European and America (0.2 - 3.0), the ratios in this work
were significantly higher and consistent with the recent studies in the NCP, which suggests that high
$ClNO_2$:$N_2O_5$ ratios were ubiquitous in the NCP and implies that the $ClNO_2$ yield via $N_2O_5$ uptake is
effective.

**4. Discussion**

**4.1 Determination of $N_2O_5$ uptake coefficients**

A composite term, $\gamma \times f$, was used to evaluate the overall $ClNO_2$ yield from $N_2O_5$ heterogeneous
hydrolysis (Mielke et al., 2013); the term was estimated by considering the production rate of $ClNO_2$
and using the measured $N_2O_5$ and $S_a$. The values calculated based on the field observations are listed
in Table S3 and had moderate variability, a range from 0.008 - 0.035 and an average of $0.019 \pm 0.009$.
Table 3 summarizes the $\gamma \times f$ values derived in the previous field observations. The value in suburban
Germany was between 0.001 and 0.09, with the average of 0.014 (Phillips et al., 2016), and the average
value in Mt. Tai, China, was approximately 0.016 (X. F. Wang et al., 2017). Therefore, the average
value in this study was comparable with that of the two suburban sites, whereas in an urban site of
Jinan, China, the value was lower than 0.008 and comparable with that in the CalNex-LA campaign.
The three sets of $\gamma \times f$ values from suburban regions were approximately twice as large as those in
urban regions, which implies that the composited $ClNO_2$ yields in the aged air masses in suburban
regions were more efficient than in the urban region. The difference of the overall yield between the
two regions may have been caused by (1) the particle morphology variation because of particle aging,
such as the particle mixing state, O:C ratio, particle viscosity and solubility (Riemer et al., 2009;
Gaston et al., 2014; Grzinic et al., 2015) or (2) the particle compound variation such as the liquid water
content and the $Cl^-$ content. The liquid water content and the $Cl^-$ content were proposed to affect the
$ClNO_2$ yield because those particle physicochemical properties were reported to affect the $N_2O_5$ uptake
coefficient (Bertram and Thornton, 2009).



According to the R4 reaction, $pNO_3^-$ and $ClNO_2$ were formed by $N_2O_5$ heterogeneous uptake, with
yields of 2 - $f$, and $f$, respectively. Following the recent work of Phillips et al., (2016), we used the
observed $pNO_3^-$ and $ClNO_2$ formation rates to derive individual $\gamma$ and $f$. The calculations assumed that
the relevant properties of the air mass are conserved and that the losses of produced species are
negligible; additionally, the $N_2O_5$ uptake coefficients and the $ClNO_2$ yield are independent of particle
size. The nights characterized by the following two features were chosen for further analysis: (1) A
clear covariance existed between the $pNO_3^-$ and $ClNO_2$, which indicated that $pNO_3^-$ and $ClNO_2$ were
to some extent predominantly produced by $N_2O_5$ uptake, and the $HNO_3$ uptake was not important for
$pNO_3^-$ formation. (2) An equivalent or increase in ammonium was accompanied by an increase of
$pNO_3^-$, which suggested that the gas-phase ammonia was repartitioned to form ammonium nitrate and
suppress the release of $HNO_3$. The rich-ammonia conditions in Beijing (Liu et al., 2017) demonstrated
that the degassing of $HNO_3$ at night can be effectively buffered by the high concentrations of ammonia
presented in the NCP. During this campaign, five nights were eligible for the following analysis. Three
different types of derivation were proposed by Phillips et al., (2016), based on the observational data
of $N_2O_5$, $ClNO_2$, $pNO_3^-$ and $S_a$; the most rigorous analysis was used in this study. The formations of
$pNO_3^-$ and $ClNO_2$ were calculated and integrated based on the measured $S_a$ and $N_2O_5$ from 5 min-
averaged datasets and an estimated initial $\gamma$ and $f$. The $\gamma$ and $f$ were optimized until good agreement
between the observed and predicted concentrations of $pNO_3^-$ and $ClNO_2$ was obtained. Figure 6 depicts
an example of the fitting results on May 28, the predicted $N_2O_5$ uptake coefficient and $ClNO2$ yield
were 0.017 and 1.0, respectively. Five sets of values of $\gamma$ and $f$ obtained are listed in Table 4. $N_2O_5$
uptake coefficients ranged from 0.012 - 0.055, with an average of 0.034 ± 0.018, and the $ClNO_2$ yield
ranged from 0.50 to unity, with an average of 0.73 ± 0.25. The errors from each derivation were 30%
- 50% and came from the field measurements of $S_a$, $N_2O_5$, $pNO_3^-$ and $ClNO_2$.
The average $\gamma$ value was consistent with the results derived by the same method in a rural site in
Germany (Phillips et al., 2016) but was higher than that found in previous studies in the UK and North
America that used different derivation methods; these methods included the steady state lifetime
method (Morgan et al., 2015; Brown et al., 2006, 2009), the iterated box model (Wagner et al., 2013)
and direct measurement based on an aerosol flow reactor (Bertram et al., 2009; Riedel et al., 2012).
The steady state lifetime method is very sensitive to $NO_2$ concentration, and since the $NO_2$
measurement suffered with ambient $NO_y$ interference, we did not apply the steady state lifetime
method in this study (Brown et al., 2003). Nonetheless, the derived $\gamma$ in Beijing showed good
agreement with the recent results derived by the steady state method in Jinan and Mt. Tai (X. F. Wang
X et al., 2017; Z. Wang et al., 2017). The consistency eliminates the discrepancy possibly brought by
the differences of analysis methods. Therefore, we suggest that fast $N_2O_5$ uptake was a ubiquitous
feature that existed in the NCP. In this study, sulfate is dominated the $PM_{1.0}$ concentration with the
percentage over 30%, which may be the reason of elevated $N_2O_5$ uptake coefficient presented in
Beijing, like the result in high sulfate air mass over Ohio and western Pennsylvania (Brown et al.,
2006). Previous studies have shown that the $N_2O_5$ uptake coefficient strongly depends on the liquid
water, the $pNO_3^-$ and organic mass; liquid water content promotes $N_2O_5$ uptake, whereas $pNO_3^-$ and
organic mass inhibit $N_2O_5$ uptake (Thornton et al., 2003, Wahner et al., 1998; McNeill et al., 2006).
Because of the limited data set of $N_2O_5$ uptake coefficients in this campaign, the trends of the
determined $N_2O_5$ uptake coefficients with the parameters mentioned above were not convincing, and
more valid data is needed for further studies of the $N_2O_5$ uptake mechanism. With respect to $f$, the





values are comparable with that observed in Germany (Phillips et al., 2016) and are similar with that
estimated in the power plant plume in Mt. Tai with high chloride content (Z. Wang et al., 2017).
**4.2 N$_2$O$_5$ lifetime and reactivity**
The lifetime of N$_2$O$_5$ was estimated by the steady state method, assuming that the production and loss
of N$_2$O$_5$ was in balance after a period following sunset. Eq. 2 for the steady state approximation has
been frequently applied in analyzing the fate of N$_2$O$_5$ (Platt et al., 1980; Allan et al., 1999; Brown et
al., 2003).
$$\tau_{ss}(N_2O_5) = \frac{1}{L_{ss}(N_2O_5)} = \frac{[N_2O_5]}{k_{NO2+O3}[NO_2][O_3]} \qquad \text{(Eq. 2)}$$
In Eq. 2 $\tau_{ss}(N_2O_5)$ denotes the steady state lifetime of N$_2$O$_5$ and $L_{ss}(N_2O_5)$ denotes the loss term of
N$_2$O$_5$ corresponding to the steady state lifetime. A numerical model was used to check the validity of
the steady state approximation (Brown et al., 2003); details are given in Figure S2. The results show
that the steady state can generally be achieved within 30 minutes. In this study, the steady state lifetime
was only calculated from 20:00 to 04:00. The time periods with NO concentration larger than 0.1 ppbv
were excluded because the steady state is easily disturbed. The overall N$_2$O$_5$ reactivity (k(N$_2$O$_5$)) can
be calculated by accumulating each individual loss term as in Eq. 3, including the N$_2$O$_5$ heterogeneous
hydrolysis and the reaction of NO$_3$ with VOCs. The NO$_3$ heterogeneous uptake and the loss of N$_2$O$_5$
via gas-phase reactions were assumed to be negligible (Brown and Stutz, 2012). $k_i$ represent the
reaction rate constants of the reaction of NO$_3$+VOCs$_i$. Isoprene and monoterpene were used in the
calculation. The N$_2$O$_5$ loss rate coefficient by heterogeneous hydrolysis was calculated by using an
average $\gamma$ of 0.034.
$$k(N_2O_5) = \frac{\sum k_i \cdot [VOCs_i]}{k_{eq} \cdot [NO_2]} + \frac{C \cdot S_a \cdot \gamma}{4} \qquad \text{(Eq. 3)}$$
The time series of the steady state lifetime of N$_2$O$_5$ is shown in Figure S3. The N$_2$O$_5$ steady state
lifetime ranged from <5 s to 1140 s, with an average of 310 ± 240 s, and large variability was shown
during the campaign. The N$_2$O$_5$ lifetimes during the BAM period were higher than those during the
UAM period, which is predictable since the clean air mass has lower N$_2$O$_5$ reactivity because of much
lower aerosol loading. Two extremely short N$_2$O$_5$ lifetime cases were captured on the nights of May
30 and June 3, with peak values below 200 s throughout those nights. Figure 7 shows that the N$_2$O$_5$
lifetime had a very clear negative dependence of the ambient aerosol surface area when larger than
300 μm$^2$ cm$^{-3}$, which indicates that the N$_2$O$_5$ heterogeneous uptake plays an important role in the
regulation of N$_2$O$_5$ lifetime. The study conducted in the residual layer of Hong Kong showed a similar
tendency despite the overall N$_2$O$_5$ lifetime being shorter at this site (Brown et al., 2016). Additionally,
a negative dependence of N$_2$O$_5$ lifetime on RH was reported in Hong Kong but was not observed in
this study (Figure S4).
Figure 8 shows the time series of the overall N$_2$O$_5$ loss rate constant as well as the steady state N$_2$O$_5$
loss rate. The overall N$_2$O$_5$ loss rate constant from the individuals was reasonably comparable with the
steady state N$_2$O$_5$ loss rate, except for the nights of 28, 30 May and 3 June, on which the $L_{ss}(N_2O_5)$





calculated by the steady state method were much higher than the overall $k(N_2O_5)$. The average $N_2O_5$
loss rate contributed by the $N_2O_5$ heterogeneous hydrolysis was $8.1\times10^{-4}$ s$^{-1}$. The average $NO_3$ loss rate
by the reaction of $NO_3$ with VOCs was $0.015 \pm 0.007$ s$^{-1}$, which is comparable with the previous results
in suburban Beijing in 2006 (H. C. Wang et al., 2017c), in which the contribution to the $N_2O_5$ reactivity
was $1.63\times10^{-3}$ s$^{-1}$. Compared with $N_2O_5$ loss via direct heterogeneous hydrolysis, the indirect loss via
$NO_3$+VOCs as dominated by approximately 67% of the total $N_2O_5$ loss. Because only a subset of the
suite of organic species at the site was measured, the calculated loss rate constant via $NO_3$+VOCs
represents a lower limit. Therefore, the $N_2O_5$ loss via $NO_3$+VOCs may occupy a larger proportion. The
overall loss rate constant from $NO_3$+VOCs and $N_2O_5$ uptake was $2.44\times10^{-3}$ s$^{-1}$, which was reasonably
lower than the steady state $N_2O_5$ loss rate constant of $3.61\times10^{-3}$ s$^{-1}$; the gap may be explained by the
unmeasured reactive VOCs or the unaccounted NO that was near the instrumental limit of detection.
**4.3 Nocturnal $NO_3$ oxidation**
Recent studies have suggested that the fate of BVOCs after sunset is dominated by $NO_X$ or $O_3$, with
variation of the ratio of $NO_X$ to BVOCs and that the nighttime oxidation is located in the transition
region between $NO_X$-domination and $O_3$-domination in the United States (Edwards et al., 2017).
During this campaign, the nocturnal average concentrations of isoprene and monoterpene were $156 \pm$
88 pptv and $86 \pm 42$ pptv, respectively. We used isoprene and monoterpene to represent a lower limit
mixing ratio of total BVOCs; the average ratio of $NO_X$/BVOC was larger than 10 and exhibited small
variation during the BAM and UAM periods. The value was much higher than the critical value
($NO_X$/BVOC = 0.5) of the transition regime proposed by Edwards et al. (2017), which suggests that
the oxidation of BVOCs in Beijing was NOx-dominated and the nighttime fate of BVOCs was
controlled by $NO_3$. Since the ONs formation via BVOC oxidation was mainly attributed to the $NO_3$
oxidation with high yield, we suggest that the ONs production capacity was maximized in the high
NOx/BVOCs region.
Similar to k(OH), the nighttime VOCs reactivity, $k(VOCs_i)$, is defined as the pseudo loss rate of
VOCs oxidized by oxidants and is expressed as Eq. 4. Here, we only consider the oxidation by $O_3$ and
$NO_3$. $k_{VOCs_i+NO_3}$ and $k_{VOCs_i+O_3}$ are the reaction constants of VOCs$_i$ with $NO_3$ and $O_3$, respectively.
$$k(VOCs_i) = k_{VOCs_i+NO_3} \cdot [NO_3] + k_{VOCs_i+O_3} \cdot [O_3] \qquad \text{(Eq. 4)}$$
During this campaign, VOCs reactivity could be determined with the measured $O_3$ and calculated $NO_3$.
Figure 9 depicts four kinds of VOCs reactivity distribution during nighttime, including the isoprene
(ISO), monoterpene (here represented by α-pinene, API), the alkenes with the double bond elsewhere
in the molecule (OLI) and the double bond at the end or terminal position of the molecule (OLT). The
reaction rates were cited from the regional atmospheric chemistry mechanism version 2 (RACM2,
Goliff et al., (2013)). The VOCs reactivity were dominated by $NO_3$ oxidation and contributed up to
90% in total; less than 10% were oxidized by $O_3$ during the nighttime. The results further confirmed
that the oxidation of BVOCs is controlled by $NO_3$ rather than $O_3$.
For calculating nocturnal ONs production from $NO_3$ oxidation of isoprene and monoterpene, as well
as the same period inorganic nitrate production via $N_2O_5$ heterogeneous uptake, the ClNO$_2$ yield was



set to the determined average value of 0.73. The organic nitrate yield of the reaction of $NO_3$ with
isoprene was set to 0.7, from Rollins et al. (2009). The yield from the reaction of $NO_3$ with
monoterpene was represented by the $NO_3 + \alpha$-pinene and was set to 0.15, following Spittler et al. (2006).
Although the yield from the $NO_3$ oxidation of isoprene is much higher than that of monoterpene, the
total ONs production was dominated by the oxidation of $NO_3$ with monoterpene because the reaction
of $NO_3$ with monoterpene is much faster than that with isoprene. Because of the lack of measurement
of alkenes and other VOCs that can react with $NO_3$ and form ONs, the calculated nighttime ONs
production rate analyzed here served as lower limit estimations. Figure 10 depicts the mean diurnal
profiles of the nocturnal formation rates of inorganic nitrates and ONs. The average production rate of
ONs was up to $0.11 \pm 0.09$ ppbv $h^{-1}$, which was much higher than that predicted in a suburban site in
Beijing in 2006, with an average value of 0.06 ppbv $h^{-1}$ (H. C. Wang et al., 2017b). In the high
$NO_X$/BVOCs air masses, the inorganic nitrate formation was proposed to increase with the increase of
sunset $NO_X$/BVOCs (Edwards et al., 2017). The formation rate of inorganic nitrate via $N_2O_5$ uptake
was significant, with an average of $0.43 \pm 0.12$ ppbv $h^{-1}$, and was much larger than the organic nitrate
formation. The NOx was mainly removed as the inorganic nitrate format by nocturnal $NO_3$-$N_2O_5$
chemistry in Beijing. Overall, the $NO_3$-$N_2O_5$ chemistry promoted significant $NO_X$ removal, with 0.54
ppbv $h^{-1}$ accounted for by the organic and inorganic nitrates, and the integral $NO_X$ removal was
approximately 5 ppbv per night. Since ONs is an important precursor of the secondary organic aerosols
(SOA), the $NO_3$ oxidation was very important from the perspective of organic aerosol formation and
regional particulate matter (e.g., Ng et al., 2008).

**5. Conclusion**
We reported an intensive field study of $NO_3$-$N_2O_5$ chemistry at a downwind suburban site in Beijing
during the summer of 2016. High levels of $ClNO_2$ and $N_2O_5$ were observed, with maxima of 2.9 ppbv
and 937 pptv (1-min), respectively. The $N_2O_5$ uptake coefficient was estimated to be in the range of
0.012-0.055, with an average value of $0.034 \pm 0.018$, and the corresponding $ClNO_2$ yield was derived
to be in the range of 0.5-1.0, with an average value of $0.73 \pm 0.25$. The elevated $ClNO_2$ levels and
$ClNO_2$/$N_2O_5$ ratios are comparable with those in chloride-rich regions in the NCP. The results highlight
fast $N_2O_5$ heterogeneous hydrolysis and efficient $ClNO_2$ formation in the outflow of urban Beijing.
Thus, its role in $O_3$ pollution in summer could be more important than in other regions.
Since the $NO_3$-$N_2O_5$ chemical equilibrium favors $NO_3$ in summer with high temperature and high
NOx, the elevated $NO_3$ dominated the nocturnal degradation of BVOCs and could lead to efficient
ONs formation. Because the air masses in Beijing featured high $NO_X$/BVOCs ratios (>10), our results
suggest that the nocturnal $NO_3$ oxidation of BVOCs was NOx-dominated. Because of the extremely
high NOx emissions, the formation of ONs may not be sensitive to the reduction of NOx but rather to
the change of unsaturated VOCs (e.g., BVOCs), which is similar to the daytime photochemical $O_3$
pollution (e.g., Lu et al., 2010) diagnosed for this area; this suggests that the control of the unsaturated
VOCs would moderate the $O_3$ pollution and ONs particulate matter in parallel. Moreover, the reduction
of NOx would also be helpful to reduce the $pNO_3^-$ formation via $N_2O_5$ heterogeneous hydrolysis under
such high NOx/BVOCs ratios (Edwards et al., 2017).





***Acknowledgements***. This work was supported by the National Natural Science Foundation of China
(Grants No. 91544225, 41375124, 21522701, 41421064, 91744204), the Strategic Priority Research
Program of the Chinese Academy of Sciences (Grants No. XDB05010500), the program on
'Photochemical smog in China" financed by the Swedish Research Council (639-2013-6917), and the
National Key R&D Program of China (Grants No. 2016YFC0202000, Task 3). The authors gratefully
acknowledge the Peking University and Gethenburg University science team for their technical support
and discussions during the Changping campaign.

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





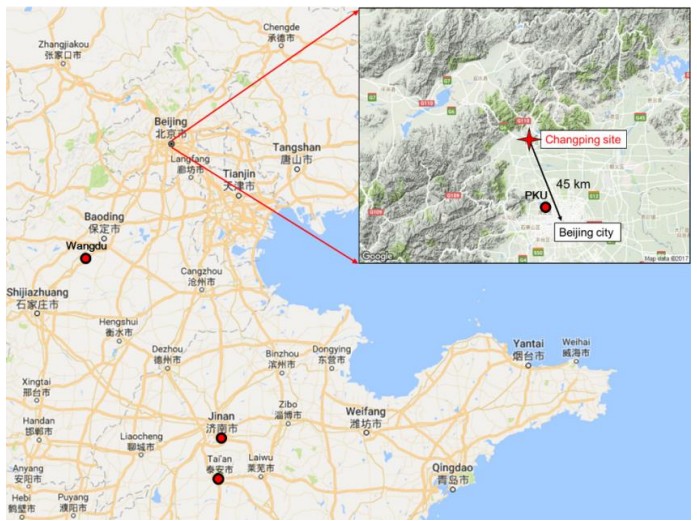


**Figure 1.** Map of Beijing and surrounding area. The red star represents the location of the Changping
site, red dots show other sites where previous $N_2O_5$ measurements were conducted in the North China
Plain (NCP), including Peking University (PKU), Wangdu, Jinan and Mt. Tai (in Tai' an).


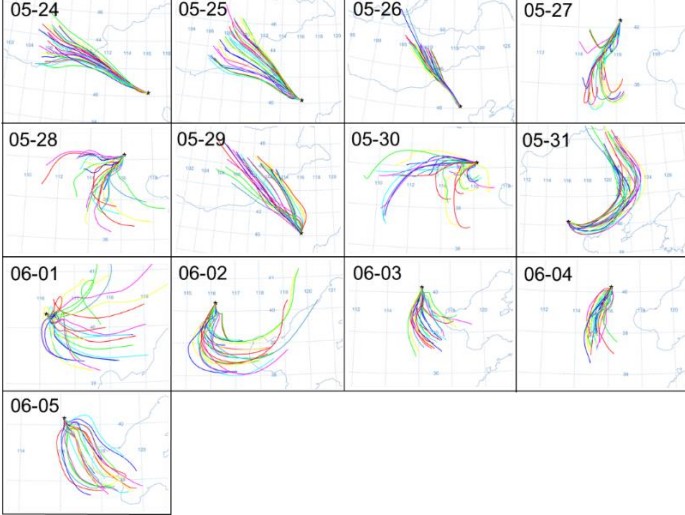


**Figure 2.** Backward trajectory calculations using the Hybrid Single-Particle Lagrangian Integrated
Trajectory (HYSPLIT) model. The images depict a 24-h history of air masses arriving at the
measurement site at 12:00 (CNST).





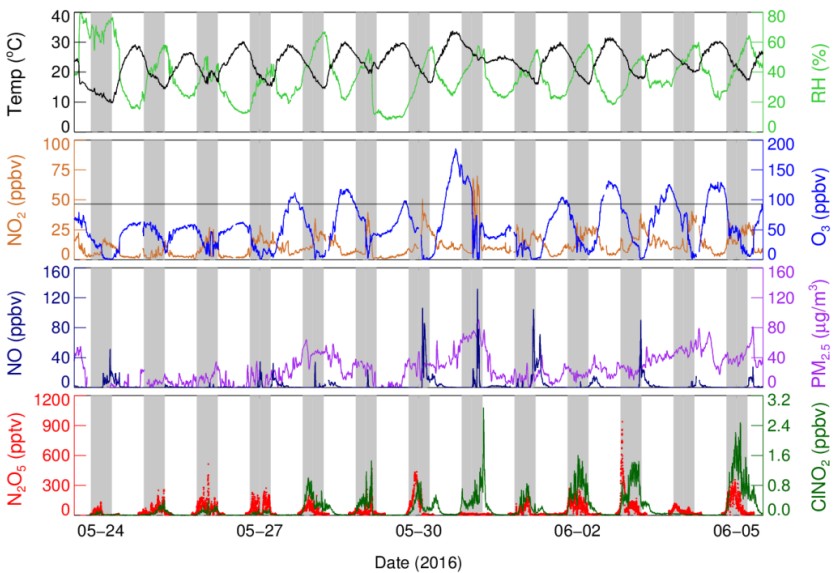


**Figure 3.** Time series of $N_2O_5$, $ClNO_2$ and other relevant parameters. The black line in the $O_3$ panel

denotes Chinese national air quality standard for $O_3$ (ca. 93 ppbv for the surface conditions).

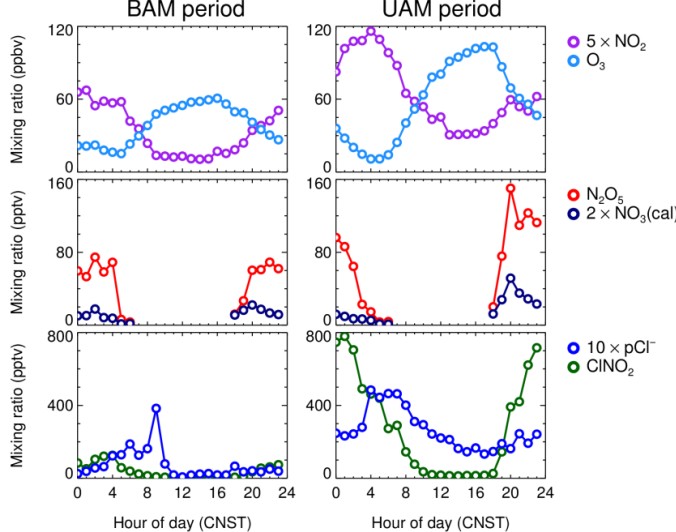


**Figure 4.** Mean diurnal profiles of $5 \times NO_2$, $O_3$, $N_2O_5$, $2 \times NO_3$ (calculated), $ClNO_2$, and $10 \times pCl^-$. The

left three panels depict the background air mass (BAM) period and the right three panels depict the

urban air mass (UAM) period.





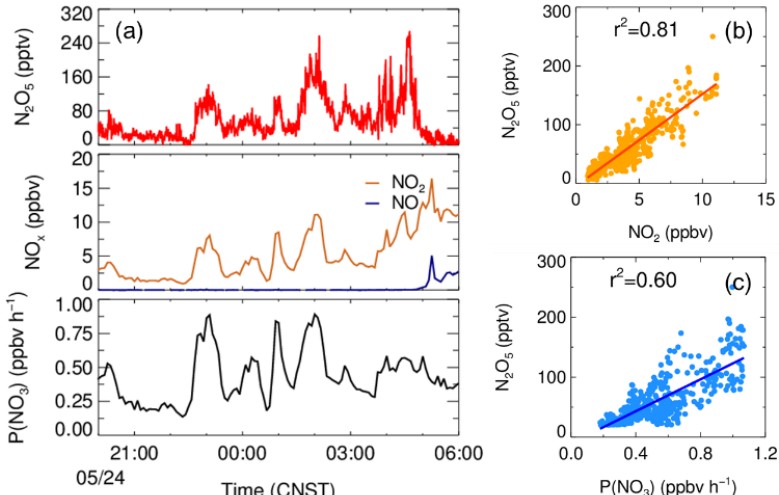


**Figure 5.** The correlation of the mixing ratio of $N_2O_5$ and $NO_2$ and the production rate of $NO_3$ on the

night of May 24.

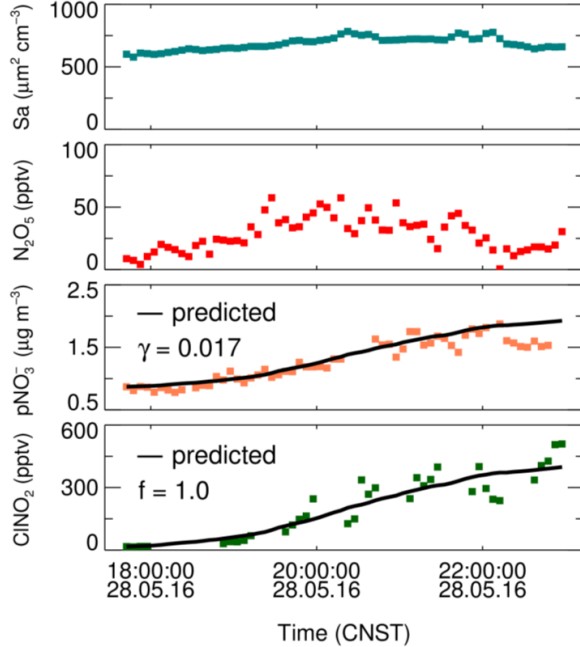


**Figure 6**. The best fitting of $\gamma$ and $f$ to reproduce the observed $ClNO_2$ and $pNO_3^-$ with an offset on May

28. The black lines are the predicted results of the integrated $pNO_3^-$ and $ClNO_2$ by using the observed
$S_a$ and $N_2O_5$.





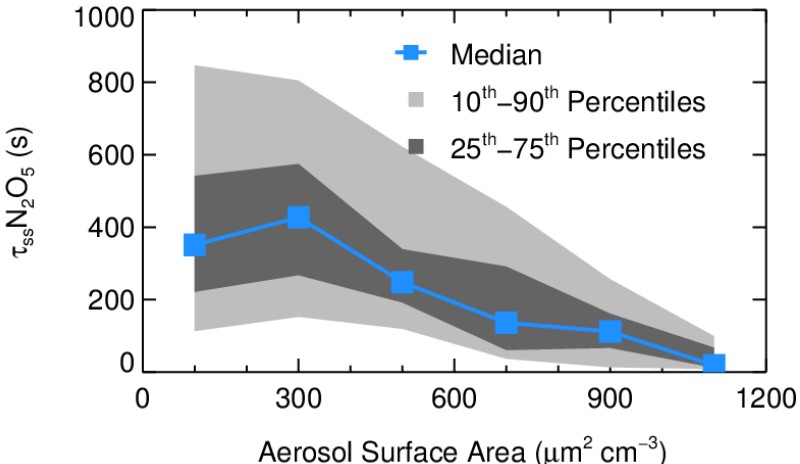


**Figure 7.** The dependence of $N_2O_5$ lifetime on aerosol surface area. Data were selected from 20:00 to 04:00 and are shown as medians, 25 - 75th percentile ranges, and 10 - 90th percentile ranges, as shown in the legend.


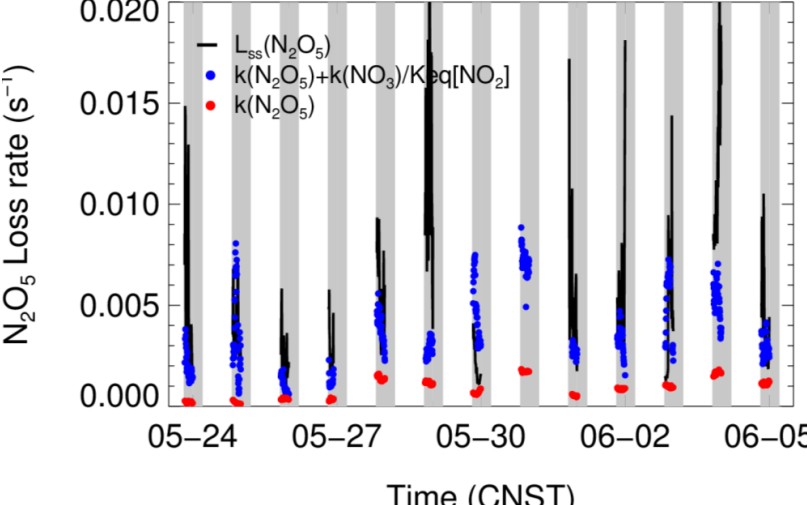


**Figure 8**. Time series of the individual $N_2O_5$ loss terms and the loss rate constant of $N_2O_5$ in steady state ($Lss(N_2O_5)$).





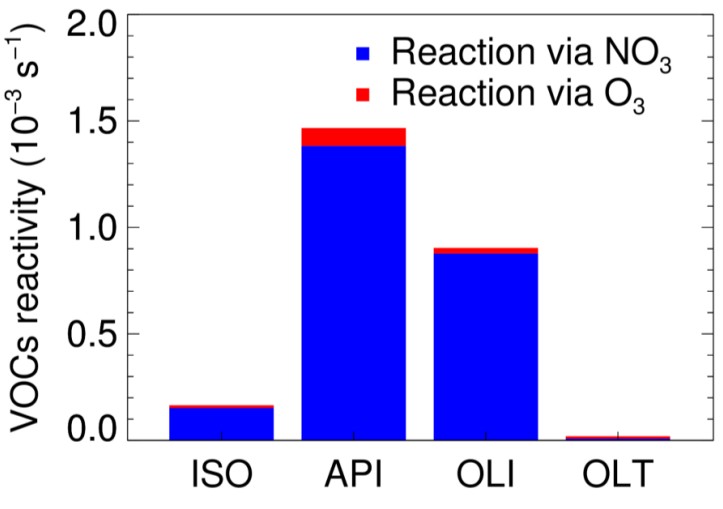

**Figure 9.** The nighttime VOCs reactivity of $NO_3$ and $O_3$; the classification was based on RACM2, and data were selected from 20:00 to 04:00.

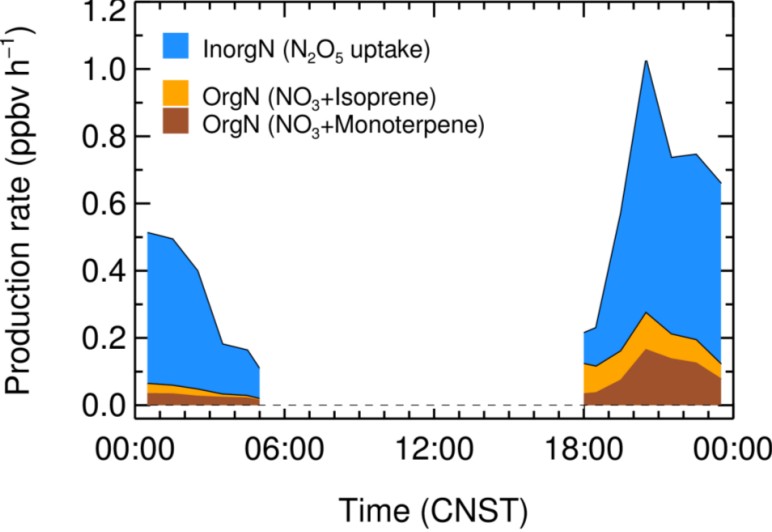

**Figure 10.** The nighttime production rate of organic and inorganic nitrates; the inorganic nitrates were calculated from the $N_2O_5$ heterogeneous hydrolysis, and the ONs were calculated by the $NO_3$ reacted with isoprene and monoterpene.



**Table 1.** The observed gas and particle parameters used in this analysis during the campaign.

| Species | Limit of detection | Methods | Accuracy |
|---|---|---|---|
| $N_2O_5$ | 2.7 pptv ($1\sigma$, 1 min) | CEAS | ± 19% |
| $ClNO_2$ | 16 pptv ($2\sigma$, 1 min) | FIGAERO-ToF-CIMS | ± 23% |
| NO | 60 pptv ($2\sigma$, 1 min) | Mo convert | ± 20% |
| $NO_2$ | 0.3 ppbv ($2\sigma$, 1 min) | Mo convert | ± 20% |
| $O_3$ | 0.5 ppbv ($2\sigma$, 1 min) | UV photometry | ± 5% |
| Aerosol surface area | -   (4 min) | SMPS, APS | ± 30% |
| VOCs | 0.1 ppbv (5 min) | PTR-MS | ± 30% |
| $PM_{2.5}$ | 0.1 $\mu g\ m^{-3}$ (1 min) | TEOM | ± 5% |
| $PM_{1.0}$ components | 0.15 $\mu g\ m^{-3}$ (4 min) | HR-ToF-AMS | ± 30% |


**Table 2.** Summary of the field observed ambient $ClNO_2/N_2O_5$.

| Location | Region | $ClNO_2/N_2O_5$ [a] | References |
|---|---|---|---|
| Beijing, China | Inland | 0.7 – 42.0 (7.7) | This work |
| Wangdu, China | Inland | 0.4 - 131.3 (29.5) | Tham et al., 2016 |
| Jinan, China | Marine | 25.0 - 118.0 [b] | Z. Wang et al., 2017 |
| Mt. Tai, China | Marine | ~ 4.0 | X. F. Wang et al., 2017 |
| Hong Kong, China | Marine | 0.1 - 2.0 | T. Wang et al., 2016 |
| London, UK | Inland | 0.02 - 2.4 (0.51) | Bannan et al., 2015 |
| Frankfurt, Germany | Inland | 0.2 - 3.0 | Phillips et al., 2012 |
| Colorado, USA | Inland | 0.2 - 3.0 | Thornton et al., 2010 |
| California, USA | Marine | ~ 0.2 - 10.0 [c] | Mielke et al., 2013 |

Note: [a] Daily average results; [b] Power plant plume cases at Mt. Tai in Shandong, China; [c] Estimated according to Mielke
et al., (2013).



**Table 3.** Summary of the average $\gamma \times f$ values derived in the field observations.

| Location | Region | $\gamma \times f$ | References |
|----------|--------|-------------------|------------|
| Beijing, China | suburban | 0.019 | This work |
| Frankfurt, Germany | suburban | 0.014 | Phillips et al., 2016 |
| Mt. Tai, China | suburban | 0.016 | X. F. Wang et al., 2017 |
| Jinan, China | urban | <0.008 | Z. Wang et al., 2017 |
| California, USA | urban | 0.008 | Mielke et al., 2013 |


**Table 4.** List of the $N_2O_5$ uptake coefficients and the yield of $ClNO_2$ in this campaign.

| Start time | End time | $\gamma$ | $f$ |
|------------|----------|----------|-----|
| 05/25 00:00 | 05/25 05:00 | 0.047 | 0.60 |
| 05/25 18:30 | 05/25 23:00 | 0.012 | 1.0 |
| 05/27 19:00 | 05/27 20:40 | 0.040 | 0.50 |
| 05/28 19:00 | 05/28 23:00 | 0.017 | 1.0 |
| 05/30 21:00 | 05/31 00:00 | 0.055 | 0.55 |
