# Peer review of "Efficient N2O5 Uptake and NO3 Oxidation in the Outflow of Urban Beijing"

_Atmospheric Chemistry and Physics, 2018_

## Referee Comment (RC1) · Anonymous Referee #1 · 9 Mar 2018

This paper presents NO3 and N2O5 observational data from a suburban site in Beijing during the summer of 2016. The authors use these data to investigate the oxidation of volatile organic compounds (VOC) by NO3 and the effect of N2O5 heterogeneous uptake on reactive nitrogen loss and ClNO2 production in the Beijing urban outflow. Nocturnal biogenic VOC oxidation was shown to be dominated by NO3, and the heterogeneous uptake of N2O5 was found to be a significant loss mechanism for reactive nitrogen. The uptake of N2O5 was found to produce approximately a factor of four more inorganic nitrate than organic nitrate from the NO3 + VOC pathway and result in significant ClNO2 production. These results are compared, and broadly agree, with previously reported observations and represent a valuable contribution to the growing body of work on the importance of nocturnal chemistry on local atmospheric composi-

tion. I recommend publication of the manuscript once the following minor comments / technical corrections have been addressed.

Minor comments / technical corrections:

Lines 60-61: Please could the authors state if these are mass or molar yields.

Line 148-150 – It would be easier for the reader if the authors could be consistent with the order of N2O5 and ClNO2 in this sentence.

Figure 3: The scale on the NO plot makes it difficult to see NO mixing ratios. Please consider either a log scale or a discontinuity to make this more visible.

Lines 276-278: This sentence is confusing, please restructure.

Line 289: Please re-reference the recent studies in the NCP.

Figure 8 and lines 389 – 392: There is an inconsistency between the text and Fig. 8. In the text the authors state that on the three days with the largest discrepancies between the steady state calculated N2O5 lifetime and that calculated using the overall k(N2O5) the steady state calculation is much higher than the overall k(N2O5). In Fig. 8 however, the discrepancy on 30th May is in the opposite direction, with the steady state lifetime approaching a factor of 2 lower than the overall k(N2O5). The authors should correct this statement and provide an explanation for this discrepancy. The authors should also explain why there is no steady state calculated N2O5 lifetime for 31st May in Fig. 8.

Figure 9: Although the acronyms used in the x-axis labels are described in the text, it would help the reader if they were re-stated in the figure caption.

Line 301 and Table 3: Please check that X. F. Wang and Z. Wang references are correct.

---

## Referee Comment (RC2) · Anonymous Referee #2 · 14 Mar 2018

Wang et al present measurements of N2O5, ClNO2 and ancillary species in the urban outflow of Beijing and thereby analyze nocturnal rates of oxidation of VOCS, NOx lifetimes and chlorine activation via heterogeneous reaction of N2O5 on chloride containing particles. N2O5 uptake coefficients were in the "usual" range and ClNO2 yields were high, implying abundant sources of chlorine. The authors use established expressions to analyze their data and the manuscript contributes to the growing literature on nighttime VOC oxidation, NOX loss and ClNO2 formation without providing significant new insight. Detracting from this work, much of the referencing seems to be an arbitrary selection (often self-citation) of related work and the estimation (or presentation) of uncertainties in derived parameters is largely missing.

The following points should be addressed (some are major) and the English language

corrected (some suggestions are listed below) before re-review.

L61 State how the yield of SOA (23.8 % or 174 %) is defined.

L70 kN2O5 is not a rate coefficient. Its best to call it a pseudo-first order loss rate constant to avoid confusing it with rate constants for gas-phase reactions

L70 Eq. (1) was certainly not derived by Tang et al in 2017. Use an appropriate (earlier) reference.

L175 The correction factor of 0.6 (independent of time of day, day of campaign, NOx, or air mass-age) is clearly a poor assumption given that the NOx to NOy ratio is highly variable in time and space. The assumption that the correction factor in Wangdu is the same as in Changping is without real basis. Note also that the photo-stationary state between NO,NO2 and O3 will break down in the presence of other oxidants (e.g. RO2) so that measurement of NO and O3 (and j-NO2) cannot replace NO2 measurements. The authors must estimate the uncertainty related to this correction factor (and thus with the NO2 measurements) is they wish to use NO2 data in any quantitative sense. This applies to section 4.2 where they calculate N2O5 lifetimes in steady state via calculation of the N2O5 production term, which requires NO2 mixing ratios. It also applies to the calculation of NO3 from the N2O5 and NO2 measurements and the equilibrium constant and this impacts on the results of section 4.2 where NO3 concentrations are used to calculated oxidation rates of VOCs. In principal, the lack of accurate NO2 measurements during this campaign reduces many conclusions of this paper to a qualitative level.

L191 "Figure 2 shows the calculated backward trajectories using the Hybrid Single-Particle Lagrangian Integrated Trajectory (HYSPLIT) model". As far as I can tell, this is the first and last mention of air-mass trajectories. I would suggest that the Figure can be relegated to the SI.

L235 "The single peak (in N2O5) occurred near 20:00 and then gradually decreased".

Is this a reproducible feature of the campaign or a bias of the mean due to one or two events. Taking the median rather than the mean would resolve this. Also, why (line 236) does the N2O5 increase before sunrise (or do the authors mean "at sunrise") ?

L243-246 "ClNO2 accumulated corresponding to N2O5 after sunset but ClNO2 peaked in the middle or the second half of the night since the nocturnal sinks of ClNO2 were negligible to our knowledge." I'm not sure what the authors are trying to say here. There are many examples that show great variability in the N2O5–to-ClNO2 ratio. The interesting part of this section (lines 243 to 267) is the discussion of the sources of chloride needed to drive the ClNO2 formation in this continental region. In principal, the chloride content of the aerosol can be calculated from the yield of ClNO2 and the appropriate expression that defines the parameter "f". I suggest the authors do this.

L249 and promoted the N2O5 conversion to ClNO2 (e.g., Roberts et al., 2009). Why this citation ? The formation of ClNO2 from N2O5 was known (and quantified) long before 2009. Cite the appropriate literature.

L279 In lines 280-290 It is not clear whether we are dealing with ratios of the concentrations of ClNO2 and N2O5 or ratios in their production rates (L282). If relative rates are calculated we need to know over which period they were derived.

L295 A composite term, $\delta$İŻ¿ $\times$ f, was used to evaluate the overall ClNO2 yield (f). . . . . .

L296 How and over what period was the production rate of ClNO2 determined ? How stable were N2O5 and Sa in this period ?

L296 the term was estimated by considering. . . .. Give the expression used to derive the composite term from the observables.

L300 and 301 (and Table 3) The average values need to be listed with standard deviations to enable comparison. The same applies to Table 4.

L313 uptake coefficients are derived from analysis of particulate nitrate and ClNO2 concentrations. Only those nights were chosen when a clear covariance between

these parameters was observed. The authors should explain how they define "clear covariance" and why, on other nights, covariance did not exist. Surely the formation of ClNO2 must always be accompanied by formation of particle nitrate ? A major issue in this analysis is the assumption that the particulate nitrate id only formed from N2O5 uptake and not influenced by (temperature dependent) HNO3 repartitioning. It appears that there were no measurements of gas-phase HNO3 or ammonia to support the contentions that this was not important. The authors must assess this rigorously and state how he uptake coefficients would be influenced by HNO3 uptake.

L327 "the most rigorous analysis was used in this study". I do not understand what this implies. Most rigorous compared to what ?

L330 Figure 6 would be improved by adding the result of a calculation with lower (factor two ?) and higher (factor two ?) uptake coefficients to test the sensitivity of the data to the derived parameter. Also, what is the source of the offset in the particle nitrate ? How does the particle nitrate look over the diel period. This is essential information when trying to understand the effects of HNO3 re-partitioning (see comment above). .

L331 "the predicted N2O5 uptake coefficient and ClNO2 yield were 0.017 and 1.0, respectively." What are the uncertainties ?

L334 "The errors from each derivation were 30% - 50% and came from the field measurements of Sa, N2O5, pNO3- and ClNO2." Using the uncertainties listed in Table 1 results in total uncertainty (propagated in quadrature) of > 50 %. I do not understand how the quoted 30-50 % was derived.

L369 "The time periods with NO concentration larger than 0.1 ppbv were excluded". Why was this threshold chosen. The lifetime of NO3 at 0.1 ppbv of NO is about 10-20 s.

L389 The uncertainties in the N2O5 loss rate need to be calculated. As this involves NO2 measurements, the uncertainty will be very large.

L403 This section deals with oxidation of VOCs and loss of NOx to nitrates (inorganic and organic). NO3 was not measured but calculated from N2O5 and NO2 (the latter also not measured properly). The NO3 concentrations derived are therefore associated with great uncertainty. This needs to be assessed and used in the subsequent discussion and comparison with O3-induced oxidation.

L416 Similar to k(OH)....... I'm not sure why OH is being mentioned here.

L422 Terpenes were measured using PTRMS, i.e. no speciation. What is the basis for assuming that ïĄą-pinene can be used as surrogate for NO3 + terpene reactivity in these air masses ?

Figure 9 Needs uncertainties on the two terms being compared.

Some (certainly not exhaustive) suggestions for improvement of the English.

L18 Nocturnal reactive nitrogen compounds play an important role in regional air pollution

L27 The concentration of the nitrate radical (NO3) was calculated assuming that....

L34 which indicates that reduction of NOx emissions cannot help reduce the nocturnal formation of ONs.

L42 NO3 can initiate the removal of many kind of anthropogenic

L58 The reactions of NO3 with several BVOCs produce considerable amounts of organic nitrates

L207 ...nocturnal nitrate radical production rate, P(NO3), was large, with an average...

L61 The reaction of NO3 with isoprene has a SOA yield of 23.8% (Ng et al., 2008). For the reaction with a monoterpene, such as limonene, the yield can reach 174% at ambient temperatures (Boyd et al., 2017).

L97 the reaction also contributed significantly to NOX.......

L259 by acid displacement

L260 "however, the photolysis with profound ClNO2 was still maintained until noon". I think the authors are trying to say that ClNO2 survived until noon ? In this context the should mention the J-values of ClNO2.

L273-276 This needs rewriting. I think the gist if this is that that the N2O5 concentration depends on the NO2 level more than on the O3 concentration. If so, please explain why.

L305 . . ..which implies that the ClNO2 formation efficiency. . ...

L403 The title of this section is misleading. NO3 is not oxidized, but the VOCs. I suggest "NO3-induced nocturnal oxidation of VOCs" or similar.

L429 For calculating nocturnal ONs production from NO3 oxidation of isoprene and monoterpene, as well as inorganic nitrate production via N2O5 heterogeneous uptake over the same period. . ..

---

## Author Comment (AC1) · 9 May 2018

**Response to Referees**

We thank the reviewers for their careful reading and their constructive comments on our manuscript. As detailed below, the reviewer's comments are shown as italicized font, our response to the comments are normal font. New or modified text is in blue.

**Referee #1**

*This paper presents $NO_3$ and $N_2O_5$ observational data from a suburban site in Beijing during the summer of 2016. The authors use these data to investigate the oxidation of volatile organic compounds (VOC) by $NO_3$ and the effect of $N_2O_5$ heterogeneous uptake on reactive nitrogen loss and $ClNO_2$ production in the Beijing urban outflow. Nocturnal biogenic VOC oxidation was shown to be dominated by $NO_3$, and the heterogeneous uptake of $N_2O_5$ was found to be a significant loss mechanism for reactive nitrogen. The uptake of $N_2O_5$ was found to produce approximately a factor of four more inorganic nitrate than organic nitrate from the $NO_3$ + VOC pathway and result in significant $ClNO_2$ production. These results are compared, and broadly agree, with previously reported observations and represent a valuable contribution to the growing body of work on the importance of nocturnal chemistry on local atmospheric composition. I recommend publication of the manuscript once the following minor comments /technical corrections have been addressed.*

Thanks for the referee's positive and helpful comments.

*Minor comments / technical corrections.*

*1. Lines 60-61: Please could the author state if these are mass or molar yields.*

These are the mass yields, and we added the explanation in the text.

Changed in line 60-61: "The reaction of $NO_3$ with isoprene has a SOA mass yield of 23.8% (Ng et al., 2008). For the reaction with monoterpene, such as limonene, the SOA mass yield can reach 174% at ambient temperatures (Boyd et al., 2017)."

*2. Line 148-150 – It would be easier for the reader if the authors could be consistent with the order of $N_2O_5$ and $ClNO_2$ in this sentence.*

Thanks for the suggestion, we now changed units of both $N_2O_5$ and $ClNO_2$ to "pptv".

*3. Figure 3: The scale on the NO plot makes it difficult to see NO mixing ratios. Please consider either a log scale or a discontinuity to make this more visible.*

We changed to log scale accordingly, and we labelled 0.06 ppbv NO in the black line in Figure 2 in the revised manuscript.

*4. Lines 276-278: This sentence is confusing, please restructure.*

We rewrote as: "The $N_2O_5$ concentration was highly correlated with $NO_2$ ($R^2 = 0.81$) and the $NO_3$ production rate ($R^2 = 0.60$), suggests the $N_2O_5$ concentration was solely response to the $NO_2$ concentration in the background air mass when enough $O_3$ is presented."

*5. Line 289: Please re-reference the recent studies in the NCP.*

We re-cited the recent studies in the NCP in Line 289 as suggested: "Tham et al., 2016; X. F. Wang et al., 2017; Z. Wang et al., 2017".

*6. Figure 8 and lines 389 – 392: There is an inconsistency between the text and Fig. 8. In the text the authors state that on the three days with the largest discrepancies between the steady state calculated $N_2O_5$ lifetime and that calculated using the overall $k(N_2O_5)$ the steady state calculation is much higher than the overall $k(N_2O_5)$. In Fig. 8 however, the discrepancy on $30^{th}$ May is in the opposite direction, with the steady state lifetime approaching a factor of 2 lower than the overall $k(N_2O_5)$. The authors should correct this statement and provide an explanation for this discrepancy. The authors should also explain why there is no steady state calculated $N_2O_5$ lifetime for $31^{st}$ May in Fig. 8.*

For the night of $29^{th}$-$30^{th}$ May (Referee quoted as $30^{th}$ May), the calculated steady state loss rate constant of $N_2O_5$ is much smaller than that of the overall $k(N_2O_5)$. We agree with the reviewer that this discrepancy needs more explanations. Considering that the large uncertainties propagated from the observed parameters (e.g., $NO_2$, $N_2O_5$, $S_a$), the discrepancies between the calculated steady state loss rate constant of $N_2O_5$ and the overall $k(N_2O_5)$ is mostly within the estimated uncertainty levels. The steady state loss rate constant on the night of $30^{th}$ May (mentioned $31^{th}$ May in the comment) was calculated in fact, but the values are much higher than 0.02, the reason of high steady state loss rate constant on the night of $30^{th}$ - $31^{th}$ May was not well

understand. In the revised manuscript, we enlarged the y-axis and changed to log scale, as well as added the error bar of the estimated uncertainties in the Figure 7.

Change in the revised manuscript:

"Figure 7 shows the time series of the overall $N_2O_5$ loss rate constant as well as the $N_2O_5$ steady state loss rate constant. The overall $N_2O_5$ loss rate constant was calculated from the individual terms (Eq.3). The uncertainties of the $N_2O_5$ steady state loss rate constant and the overall $k(N_2O_5)$ are estimated to be 67% and 95%, respectively (Eq. 7 and Eq. 8). The largest error sources were from the corrected $NO_2$ measurements so that it is really important to have accurate $NO_2$ measurement instrument involved in the future campaigns.

$$\frac{\Delta Lss(N_2O_5)}{Lss(N_2O_5)} = \sqrt{(\frac{\Delta[N_2O_5]}{[N_2O_5]})^2 + (\frac{\Delta[NO_2]}{[NO_2]})^2 + (\frac{\Delta[O_3]}{[O_3]})^2 + (\frac{\Delta K_{eq}}{K_{eq}})^2} \qquad (\text{Eq. 7})$$

$$\frac{\Delta k(N_2O_5)}{k(N_2O_5)} = \sqrt{(\frac{\Delta[N_2O_5]}{[N_2O_5]})^2 + (\frac{\Delta[S_a]}{[S_a]})^2 + (\frac{\Delta[\gamma]}{[\gamma]})^2 + (\frac{\Delta[NO_2]}{[NO_2]})^2 + (\frac{\Delta[O_3]}{[O_3]})^2 + (\frac{\Delta[VOC_s]}{[VOC_s]})^2 + (\frac{\Delta K_{eq}}{K_{eq}})^2} \qquad (\text{Eq. 8})$$

On the night of 29 May, the steady state loss rate constant was much lower than the overall $k(N_2O_5)$; on the nights of 28, May and 3 June, the $L_{ss}(N_2O_5)$ calculated by the steady state method were much higher than the overall $k(N_2O_5)$, but these discrepancies were in the range of the uncertainties. Except the case happened on the night of 30 May, when the steady state loss rate constant was about ten times higher than the overall loss rate constant, and the reason was not well understood according to the available parameters that we have detected. In general, the overall $N_2O_5$ loss rate constant and the steady state $N_2O_5$ loss rate constant were comparable taking into considerations of the uncertainties. "

[Figure]

**Figure 7**. Time series of the individual $N_2O_5$ loss terms and the loss rate constant of $N_2O_5$ in steady state ($L_{ss}(N_2O_5)$).

*7. Figure 9: Although the acronyms used in the x-axis labels are described in the text, it would help the reader if they were re-stated in the figure caption.*

Added the description in the caption of Figure. 9 as following: "The nighttime VOCs reactivity of $NO_3$ and $O_3$ (defined as the pseudo first order loss rate of VOCs initialed by oxidants, include $NO_3$ and $O_3$); the VOCs are classified as isoprene (ISO), monoterpene (MNT), the terminal alkenes (OLT) and the internal alkenes (OLI). The data were selected from 20:00 to the next day 04:00."

*8. Line 301 and Table 3: Please check that X. F. Wang and Z. Wang references are correct.*

We corrected the reference accordingly. The study conducted in Mt. Tai, China was from Z. Wang et al., 2017 and the study conducted in Jinan, China was from Z. Wang et al., 2017

---

## Author Comment (AC2) · 9 May 2018

**Response to Referees**

We thank the reviewers for their careful reading and their constructive comments on our manuscript. As detailed below, the reviewer's comments are shown as italicized font, our response to the comments are normal font. New or modified text is in blue.

**Referee #2**

*Wang et al present measurements of $N_2O_5$, $ClNO_2$ and ancillary species in the urban outflow of Beijing and thereby analyze nocturnal rates of oxidation of VOCS, NOx lifetimes and chlorine activation via heterogeneous reaction of $N_2O_5$ on chloride containing particles. $N_2O_5$ uptake coefficients were in the "usual" range and $ClNO_2$ yields were high, implying abundant sources of chlorine. The authors use established expressions to analyze their data and the manuscript contributes to the growing literature on nighttime VOC oxidation, $NO_x$ loss and $ClNO_2$ formation without providing significant new insight. Detracting from this work, much of the referencing seems to be an arbitrary selection (often self-citation) of related work and the estimation (or presentation) of uncertainties in derived parameters is largely missing. The following points should be addressed (some are major) and the English language corrected (some suggestions are listed below) before re-review.*

Thanks for the referee's careful and constructive comments. We checked and cited the references carefully in the revised manuscript. The uncertainties analysis was added as suggested.

*1. L61 State how the yield of SOA (23.8 % or 174 %) is defined.*

These are the mass yields, and we revised accordingly in the text. Change in the revised text: "The reaction of $NO_3$ with isoprene has a SOA mass yield of 23.8% (Ng et al., 2008). For the reaction with a monoterpene, such as limonene, the SOA mass yield can reach 174% at ambient temperatures (Boyd et al., 2017)."

*2. L70 $kN_2O_5$ is not a rate coefficient. Its best to call it a pseudo-first order loss rate constant to avoid confusing it with rate constants for gas-phase reactions.*

We change accordingly.

*3. L70 Eq. (1) was certainly not derived by Tang et al in 2017. Use an appropriate (earlier) reference.*

We cited the reference: "Wahner et al., 1998".

*4. L175 The correction factor of 0.6 (independent of time of day, day of campaign, $NO_x$, or air mass-age) is clearly a poor assumption given that the $NO_x$ to $NO_y$ ratio is highly variable in time and space. The assumption that the correction factor in Wangdu is the same as in Changping is without real basis. Note also that the photo-stationary state between NO, $NO_2$ and $O_3$ will break down in the presence of other oxidants (e.g. $RO_2$) so that measurement of NO and $O_3$ (and j-$NO_2$) cannot replace $NO_2$ measurements. The authors must estimate the uncertainty related to this correction factor (and thus with the $NO_2$ measurements) is they wish to use $NO_2$ data in any quantitative sense. This applies to section 4.2 where they calculate $N_2O_5$ lifetimes in steady state via calculation of the $N_2O_5$ production term, which requires $NO_2$ mixing ratios. It also applies to the calculation of $NO_3$ from the $N_2O_5$ and $NO_2$ measurements and the equilibrium constant and this impacts on the results of section 4.2 where $NO_3$ concentrations are used to calculated oxidation rates of VOCs. In principal, the lack of accurate $NO_2$ measurements during this campaign reduces many conclusions of this paper to a qualitative level.*

According to the reviewer's suggestions, we now extensively evaluated the influence of the uncertainty of the used $NO_2$ concentrations on the deduced VOCs (+$NO_3$) and $N_2O_5$ reactivity.

Line 177: "The correction factor (0.6) used to be the averaged scaled value of the correction factors during nighttime, the standard deviation of the daytime correction factor for all the air masses experienced at Changping site was determined to be 0.27 (1σ), which extended to nighttime and result in an uncertainty of correction to be 45%. The uncertainty of $NO_2$ is therefore about 50% when further included the associated measurement uncertainty from calibrations."

According to a Gaussian error propagation approach (see the following equations), the uncertainties of the calculated steady state lifetime, the overall k($N_2O_5$) and the $NO_3$ concentration were determined to be 67%, 95% and 67%, respectively.

We revised the paper correspondingly as follows:

Firstly, changed in line 242: "the uncertainty of $NO_3$ calculation was estimated to be 67% according to Eq. 2 which is dominated by uncertainty of the $NO_2$ concentrations.

$$\frac{\Delta[NO_3]}{[NO_3]} = \sqrt{(\frac{\Delta[N_2O_5]}{[N_2O_5]})^2 + (\frac{\Delta[NO_2]}{[NO_2]})^2 + (\frac{\Delta[O_3]}{[O_3]})^2 + (\frac{\Delta K_{eq}}{K_{eq}})^2} \qquad \text{(Eq. 2)}$$ "

Secondly, the $N_2O_5$ loss rate constant was revised in Figure 7, the error bar was added to denote the uncertainties of $N_2O_5$ steady state loss constant and the overall $N_2O_5$ loss rate constant (as $NO_2$ concentration affected the contribution of $NO_3$ oxidation).

Changed in line 390: "Figure 7 shows the time series of the overall $N_2O_5$ loss rate constant as well as the $N_2O_5$ steady state loss rate constant. The overall $N_2O_5$ loss rate constant was calculated from the individual terms (Eq.3). The uncertainties of the $N_2O_5$ steady state loss rate constant and the overall $k(N_2O_5)$ are estimated to be 67% and 95%, respectively (Eq. 7 and Eq. 8). The largest error sources were from the corrected $NO_2$ measurements so that it is really important to have accurate $NO_2$ measurement instrument involved in the future campaigns.

$$\frac{\Delta Lss(N_2O_5)}{Lss(N_2O_5)} = \sqrt{(\frac{\Delta[N_2O_5]}{[N_2O_5]})^2 + (\frac{\Delta[NO_2]}{[NO_2]})^2 + (\frac{\Delta[O_3]}{[O_3]})^2 + (\frac{\Delta K_{eq}}{K_{eq}})^2} \qquad \text{(Eq. 7)}$$

$$\frac{\Delta k(N_2O_5)}{k(N_2O_5)} = \sqrt{(\frac{\Delta[N_2O_5]}{[N_2O_5]})^2 + (\frac{\Delta[S_a]}{[S_a]})^2 + (\frac{\Delta[\gamma]}{[\gamma]})^2 + (\frac{\Delta[NO_2]}{[NO_2]})^2 + (\frac{\Delta[O_3]}{[O_3]})^2 + (\frac{\Delta[VOC_s]}{[VOC_s]})^2 + (\frac{\Delta K_{eq}}{K_{eq}})^2} \qquad \text{(Eq. 8)}$$

[Figure]

Figure 7. Time series of the individual $N_2O_5$ loss terms and the loss rate constant of $N_2O_5$ in steady state ($Lss(N_2O_5)$).

Thirdly, the uncertainty of VOCs loss rate by $NO_3$ was added in the Figure 8. Changed in line 422: "Previous measurement indicated the main detectable monoterpenes were α-pinene and β-pinene in summer Beijing (personal communication with Ying Liu). Here we assumed $\alpha$-pinene and $\beta$-pinene contributes

equally to the mixing ratios of the monoterpenes. The average value of the rate coefficients of $\alpha$-pinene and $\beta$-pinene with $NO_3$ (Atkinson and Arey, 2003) was used as the rate coefficient of monoterpene with $NO_3$. The uncertainty of the monoterpene + $NO_3$ rate coefficient in these air masses is thus estimated to be 50%. Since the uncertainty of calculated $NO_3$ is estimated to be 67%, the overall uncertainty of monoterpene reactivity toward $NO_3$ was calculated to be 85% according to a Gaussian propagation method, the uncertainties of other VOCs reactivity toward $NO_3$ was calculated to be 75% by assuming the uncertainties of the corresponding bimolecular rate constants to be 30%."

[Figure]

Figure 8. The nighttime VOCs reactivity of $NO_3$ and $O_3$ (defined as the pseudo first order loss rate of VOCs initialed by oxidants, include $NO_3$ and $O_3$); the VOCs classified as isoprene (ISO), monoterpene (MNT), the terminal alkenes (OLT) and the internal alkenes (OLI). The data were selected from 20:00 to the next day 04:00.

*5. L191 "Figure 2 shows the calculated backward trajectories using the Hybrid Single Particle Lagrangian Integrated Trajectory (HYSPLIT) model". As far as I can tell, this is the first and last mention of air-mass trajectories. I would suggest that the Figure can be relegated to the SI.*

We changed accordingly.

*6. L235 "The single peak (in $N_2O_5$) occurred near 20:00 and then gradually decreased". Is this a reproducible feature of the campaign or a bias of the mean due to one or two events. Taking the median rather than the mean would resolve this. Also, why (line 236) does the $N_2O_5$ increase before sunrise (or do the authors mean "at sunrise")?*

Thanks for the suggestion, we checked the median value of $N_2O_5$ and $NO_3$, the peak also occurred near 20:00. Therefore, we rewrote the description as following: "A peak occurred near 20:00 and decreased below the instrument detection limit at sunrise". We corrected to "at sunrise" in Line 236 accordingly.

*7. L243-246 "ClNO₂ accumulated corresponding to N₂O₅ after sunset but ClNO₂ peaked in the middle or the second half of the night since the nocturnal sinks of ClNO₂ were negligible to our knowledge." I'm not sure what the authors are trying to say here.*

We rewrote the sentence as following: "The observed ClNO₂ concentrations showed a clear increase after sunset and reached a maximum before sunrise for BAM period while reached a maximum around midnight for the UAM period."

*There are many examples that show great variability in the N₂O₅–to-ClNO₂ ratio. The interesting part of this section (lines 243 to 267) is the discussion of the sources of chloride needed to drive the ClNO₂ formation in this continental region. In principal, the chloride content of the aerosol can be calculated from the yield of ClNO₂ and the appropriate expression that defines the parameter "f". I suggest the authors do this.*

Thanks for the suggestion. We added the following discussion in the revised text: "The required nocturnal source of Cl⁻ to support the ClNO₂ production is further estimated through its loss rate. The $\gamma \times f$ was set to the campaign average value (0.019) (see Sect. 4.1), and real-time Cl⁻ loss rate via N₂O₅ can be calculated based on the measured N₂O₅ and Sa by Eq.3.

$$L[Cl^-] = (\gamma \times f) \cdot \int_{t_{sunset}}^{t_{sunrise}} \frac{C \cdot S_a}{4} [N_2O_5]dt \qquad \text{(Eq. 3)}$$

Here the $L(Cl^-)$ denotes the integral Cl⁻ loss to form the ClNO₂ per night. The required source term of the Cl⁻ need to support the ClNO₂ formation during the campaign was range from (0.5 - 4.0 ppbv per night) with (1.7 ± 2.3 ppbv per night) on average. The gas phase HCl predicted by the ISORROPIA II model showed that the HCl concentration near sunset period was high enough (much larger than 2 ppbv) to support the ClNO₂ formation (Figure. S3).

[Figure]

Figure S3. The predicted gas phase HCl concentrations by ISORRPIA II model."

*L249 and promoted the N₂O₅ conversion to ClNO₂ (e.g., Roberts et al., 2009). Why this citation? The formation of ClNO₂ from N₂O₅ was known (and quantified) long before 2009. Cite the appropriate literature.*

Corrected the citation as following: "Finlayson-Pitts et al., 1989; Behnke et al., 1997"

*8. L279 In lines 280-290. It is not clear whether we are dealing with ratios of the concentrations of ClNO₂ and N₂O₅ or ratios in their production rates (L282). If relative rates are calculated we need to know over which period they were derived.*

The daily average or median ratio of the mixing ratio of $ClNO_2$ to $N_2O_5$ was calculated from 20:00 to the next day 04:00, and the ratio of their production rates was not calculated here.

Revised the description as following: "We used the concentration ratio of $ClNO_2$ to $N_2O_5$, to describe the conversion capacity of $N_2O_5$ to $ClNO_2$. The nighttime peak values and mean values of $ClNO_2$: $N_2O_5$ were used to calculate the ratios are listed in Table S2, the calculation period is from 19:30 to the next day 05:00."

*9. L295 A composite term, γ× f, was used to evaluate the overall ClNO₂ yield (f). . . . .*

The sentence was rewrote as following: "A composite term, $\gamma \times f$, was used to evaluate the production of $ClNO_2$ from $N_2O_5$ heterogeneous hydrolysis (Mielke et al., 2013)"

*10. L296 How and over what period was the production rate of ClNO₂ determined? How stable were N₂O₅ and Sₐ in this period? L296 the term was estimated by considering. Give the expression used to derive the composite term from the observables.*

In the revised paper, we added the expression and the corresponding explanation to derive the composite term, $\gamma \times f$, as the following:

"The term, $\gamma \times f$, was estimated by fitting the observed $ClNO_2$ in a time period when the nighttime concentrations of $ClNO_2$ kept increasing. The increased $ClNO_2$ was assumed to be solely from the $N_2O_5$ uptake. The fitting was optimized by changing the input of $\gamma \times f$ associated with the measured $N_2O_5$ and $S_a$, until the $ClNO_2$ increasing was well reproduced (Eq. 4). Here $t_0$ and $t$ denote the start time and end

time, respectively, $[ClNO_2](t_0)$ is the observed concentration at $t_0$ and set as the fitting offset. The calculation time duration was normally several hours, and the derived $\gamma \times f$ was found to be constant with small uncertainties for optimization (see Table S3) (e.g., a case showed in the following Figure A1). It is worth to be noticed that both the $N_2O_5$ and $S_a$ is not necessary to be stable in this calculation due to the use of integration.

$$[ClNO_2](t) = [ClNO_2](t_0) + (\gamma \times f) \cdot \int_{t_0}^{t} \frac{C \cdot S_a}{4} [N_2O_5] dt \qquad \text{(Eq. 4)}$$

Figure A1. The reproduction of $ClNO_2$ by observed $N_2O_5$ and $S_a$.

*11. L300 and 301 (and Table 3) the average values need to be listed with standard deviations to enable comparison. The same applies to Table 4.*

The standard deviation of this study was added in both Table 3 and Table 4.

*12. L313 uptake coefficients are derived from analysis of particulate nitrate and $ClNO_2$ concentrations. Only those nights were chosen when a clear covariance between these parameters was observed. The authors should explain how they define "clear covariance" and why, on other nights, covariance did not exist.*

Here the "clear covariance" is pointing to the conditions when the square of the correlation coefficient is larger than 0.5 ($R^2 > 0.5$). Changed in line 313: "For some nights, significant correlations between $pNO_3^-$ and $ClNO_2$ were presented ($R^2 > 0.5$); while on the other nights, the $R^2$ were always smaller than 0.2, which is not meet the theoretical hypothesis of this method. In this case, we chose the nights with high correlations."

The reasons for the significant different correlations presented between the two groups of nights are still unclear. We did not find any observed parameters to explain the difference.

*Surely the formation of ClNO₂ must always be accompanied by formation of particle nitrate? A major issue in this analysis is the assumption that the particulate nitrate is only formed from N₂O₅ uptake and not influenced by (temperature dependent) HNO₃ repartitioning. It appears that there were no measurements of gas-phase HNO₃ or ammonia to support the contentions that this was not important. The authors must assess this rigorously and state how the uptake coefficients would be influenced by HNO₃ uptake.*

Unfortunately, we did not have the gas-phase $HNO_3$ or ammonia during this campaign. Our deduction on this point is as the follows,

Firstly, the daytime produced $HNO_3$ will soon be in equilibrium with the particulate nitrate within a time scale of about hundred seconds so that the daytime influence will be removed at the very beginning at night (cf. Figure A2, the observations of $HNO_3$ at summer Beijing in 2015).

[Figure]

Figure A2. The mean diurnal variation of $HNO_3$ during a campaign conducted in June 2016 in urban Beijing.

Secondly, we think the nighttime production of $HNO_3$ is very small mainly due to the small nighttime OH concentrations. Since the available nighttime OH measurements were still under big discussions (e.g., Tan et al., ACP, 2017), we think the nighttime production of $HNO_3$ from $OH+NO_2$ can be neglected according to the modeled OH concentrations (of about $1\times10^5$ $cm^{-3}$). Nevertheless, the unknown nighttime OH chemistry and the possible nighttime produced HNO3 sheds an

uncertainty on our current analysis. The impact will be the possible overestimation of the uptake coefficient of $N_2O_5$ in the current analysis framework. We now extensively discussed the possible influence of the nighttime production of $HNO_3$ and repartitioning in the revised text as: "The daytime produced $HNO_3$ will be soon in a new equilibrium with the particulate nitrate within a time scale of about hundred seconds; the nighttime source of $HNO_3$ are normally negligible except there are significant unknown OH sources at night. Both the gas-particle repartitioning of $HNO_3$ and nighttime produced $HNO_3$ will result in the overestimation of $\gamma$ and underestimation of $f$."

Reference: Tan, Z., Fuchs, H., Lu, K., Hofzumahaus, A., Bohn, B., Broch, S., Dong, H., Gomm, S., Häseler, R., He, L., Holland, F., Li, X., Liu, Y., Lu, S., Rohrer, F., Shao, M., Wang, B., Wang, M., Wu, Y., Zeng, L., Zhang, Y., Wahner, A., and Zhang, Y.: Radical chemistry at a rural site (Wangdu) in the North China Plain: observation and model calculations of OH, $HO_2$ and $RO_2$ radicals, Atmos. Chem. Phys., 17, 663-690, 10.5194/acp-17-663-2017, 2017.

*13. L327 "the most rigorous analysis was used in this study". I do not understand what this implies. Most rigorous compared to what?*

In Phillips et al., (2016), the first and simplest method is to derive $f$ only by using longer time periods (several hours or the whole night) where plots of $ClNO_2$ and $NO_3^-$ are approximately linear. The second method is to calculate absolute production rates of $NO_3^-$ and $ClNO_2$ in shorter periods (1-3 h), when $NO_3^-$ and $ClNO_2$ concentrations both increase during a period of relatively constant composition and environmental variables, such as temperature and RH. In this case, values of $pClNO_2$ and $pNO_3^-$ and average values of $S_a$ and $N_2O_5$ are used to derive $\gamma$ and $f$. The last and rigorous method is to avoid the use of the averaged $S_a$ and $N_2O_5$ in the calculation, the measured $N_2O_5$, $ClNO_2$, $S_a$, R, T and $NO_3^-$ were used directly in the calculation in a way of integration (the time step of the calculation were chose to be as small as possible, i.e., the time resolution of the associated measurement parameters). In this study, we used the last method to calculate the $N_2O_5$ uptake and $ClNO_2$ yield. In the revised manuscript, we changed the description and rewrote this part in line 349 as following: "Based on the observational data of $N_2O_5$, $ClNO_2$, $pNO_3^-$ and $S_a$ with the time resolution of 5 minutes, the formations of $pNO_3^-$ and $ClNO_2$ were calculated and integrated to reproduce the increasing of $pNO_3^-$ and $ClNO_2$ with estimated values for $\gamma$ and $f$. The offset of particle nitrate and $ClNO_2$ is the measured particle nitrate and $ClNO_2$ concentration at the start time. The $\gamma$ and $f$ were optimized based on the

Levenberg-Marquardt algorithm until good agreement between the observed and predicted concentrations of $pNO_3^-$ and $ClNO_2$ was obtained (Phillips et al., 2016)."

*14. L330 Figure 6 would be improved by adding the result of a calculation with lower (factor two?) and higher (factor two?) uptake coefficients to test the sensitivity of the data to the derived parameter. Also, what is the source of the offset in the particle nitrate? How does the particle nitrate look over the diel period? This is essential information when trying to understand the effects of HNO₃ re-partitioning (see comment above).*

Thanks for the suggestion, we estimated that the uncertainty of the determined $N_2O_5$ uptake coefficient was about 55% - 100% (55% shows below as Figure 5), and the scatter of the observed data points could then be explained by the uncertainty of the uptake coefficients. The offset of particle nitrate and $ClNO_2$ is the measured particle nitrate and $ClNO_2$ concentration at the start time point. Normally, the calculation period was the particle nitrate with increasing tendency. We checked the mean diurnal variation of particle nitrate (shows in the Figure A3), which is increased throughout the whole night and continued to the midday. The change of the particulate nitrate is not always follow the re-partitioning due to the temperature change. Nevertheless, we deduced that the impact of HNO₃ re-partitioning shall be small at night as presented in our answer to comment 12.

Changed in line 351: "The offset of particle nitrate and $ClNO_2$ is the measured particle nitrate and $ClNO_2$ concentration at the start time."

[Figure]

Figure 5. The best fit of $\gamma$ and $f$ to reproduce the observed $ClNO_2$ and $pNO_3^-$ with an offset on May 28. The black lines are the predicted results of the integrated $NO_3^-$ and $ClNO_2$ by using the observed $S_a$ and $N_2O_5$.

[Figure]

Figure A3. The mean diurnal variation of particulate nitrate during the campaign.

*15. L331 "the predicted $N_2O_5$ uptake coefficient and $ClNO_2$ yield were 0.017 and 1.0, respectively." What are the uncertainties?*

The uncertainties added in Table 4, and we added the description: "The uncertainty on each individual fitting is varied from 55% - 100% due to the variability and measurements uncertainties of $pNO_3^-$ and $ClNO_2$."

*16. L334 "The errors from each derivation were 30% - 50% and came from the field measurements of S$_a$, N$_2$O$_5$, pNO$_3^-$ and ClNO$_2$." Using the uncertainties listed in Table 1 results in total uncertainty (propagated in quadrature) of > 50 %. I do not understand how the quoted 30-50 % was derived.*

As suggested, the propagated uncertainty was added up to 55% according to a Gaussian error propagation approach, here we corrected to "approximately 55%".

*17. L369 "The time periods with NO concentration larger than 0.1 ppbv were excluded". Why was this threshold chosen? The lifetime of NO$_3$ at 0.1 ppbv of NO is about 10-20 s.*

The data selection through NO concentrations is based on the assumption that the observed NO smaller than 0.1 ppbv are very small (close to zero). This assumption is plausible as shown by the following analysis. According to a histogram analysis of the observed NO and O$_3$ concentrations for the conditions of NO smaller than 0.1 ppbv (see the following figure A3), the O$_3$ concentrations are always larger than 10 ppbv and the NO concentrations are nicely fitting to the Gaussian Distribution, suggesting most of the NO concentration below 0.1 ppbv are instrument noise and the actual value shall be very close to zero. For more rigorous analysis, we constrain the NO concentration of 0.06 ppbv (instrument LOD) in the steady state analysis of the revised text.

[Figure]

Figure A3. The histogram plot of measured NO concentration below 0.1 ppbv.

Changed line 369: "In this study, the steady state lifetime was only calculated from 20:00 to the next day 04:00. The time periods with NO concentration larger than 0.06 ppbv (instrument LOD) were excluded because the steady state is easily disturbed."

Changed line 378: "The $N_2O_5$ steady state lifetime ranged from <5 s to 1260 s, with an average of 270 ± 240 s, and large variability was shown during the campaign."

*18. L389 the uncertainties in the $N_2O_5$ loss rate need to be calculated. As this involves $NO_2$ measurements, the uncertainty will be very large.*

Added the following description in the revised text: "Figure 7 shows the time series of the overall $N_2O_5$ loss rate constant as well as the $N_2O_5$ steady state loss rate constant. The overall $N_2O_5$ loss rate constant was calculated from the individual terms (Eq.3). The uncertainties of the $N_2O_5$ steady state loss rate constant, the overall $k(N_2O_5)$ are estimated to be 67% and 95%, respectively (Eq. 7 and Eq. 8). The largest error sources were from the corrected $NO_2$ measurements so that it is really important to have accurate $NO_2$ measurement instrument involved in the future campaigns.

$$\frac{\Delta Lss(N_2O_5)}{Lss(N_2O_5)} = \sqrt{\left(\frac{\Delta[N_2O_5]}{[N_2O_5]}\right)^2 + \left(\frac{\Delta[NO_2]}{[NO_2]}\right)^2 + \left(\frac{\Delta[O_3]}{[O_3]}\right)^2 + \left(\frac{\Delta K_{eq}}{K_{eq}}\right)^2} \qquad \text{(Eq. 7)}$$

$$\frac{\Delta k(N_2O_5)}{k(N_2O_5)} = \sqrt{\left(\frac{\Delta[N_2O_5]}{[N_2O_5]}\right)^2 + \left(\frac{\Delta[S_a]}{[S_a]}\right)^2 + \left(\frac{\Delta[\gamma]}{[\gamma]}\right)^2 + \left(\frac{\Delta[NO_2]}{[NO_2]}\right)^2 + \left(\frac{\Delta[O_3]}{[O_3]}\right)^2 + \left(\frac{\Delta[VOC_s]}{[VOC_s]}\right)^2 + \left(\frac{\Delta K_{eq}}{K_{eq}}\right)^2} \qquad \text{(Eq. 8)."}$$

*19. L403 This section deals with oxidation of VOCs and loss of NOx to nitrates (inorganic and organic). $NO_3$ was not measured but calculated from $N_2O_5$ and $NO_2$ (the latter also not measured properly). The $NO_3$ concentrations derived are therefore associated with great uncertainty. This needs to be assessed and used in the subsequent discussion and comparison with $O_3$-induced oxidation.*

We carefully performed the uncertainty analysis of the calculated $NO_3$ concentrations as suggested. We found the uncertainty of calculated $NO_3$ is 67% associated with the uncertainties of $NO_2$ and $N_2O_5$. We also added the following description in the revised text. Added in line 403: "Even the $NO_3$ concentration in the lower range, $NO_3$ still responsible for more than 70% nocturnal BVOCs oxidation. The results further confirmed that the oxidation of BVOCs is controlled by $NO_3$ rather than $O_3$ in summer Beijing."

*20. L416 Similar to k(OH). . .. . . I'm not sure why OH is being mentioned here.*

Deleted the "Similar to $k$(OH),".

*21. L422 Terpenes were measured using PTRMS, i.e. no speciation. What is the basis*

*for assuming that alpha-pinene can be used as surrogate for NO₃ + terpene reactivity in these air masses?*

The speciation measurements of monoterpene are still quite sparse in China. We have now discussed with an expert on this topic. We learnt that the major monoterpene species in Summer Beijing were α-pinene and β-pinene according to GC-MS measurements.

Changed in line 422: "Previous measurement indicated the main detectable monoterpenes were α-pinene and β-pinene in summer Beijing (personal communication with Ying Liu). Here we assumed *α*-pinene and *β*-pinene contributes equally to the mixing ratios of the monoterpenes. The average value of the rate coefficients of *α*-pinene and *β*-pinene with $NO_3$ (Atkinson and Arey, 2003) was used as the rate coefficient of monoterpene with $NO_3$. The uncertainty of the monoterpene + $NO_3$ rate coefficient in these air masses is thus estimated to be 50%."

*22. Figure 9 Needs uncertainties on the two terms being compared.*

Thanks for the suggestion, we added the error bar in the Figure 9, the uncertainty of $NO_3$ calculation initialed by $NO_2$ was discussed in Question NO. 4

*23. Some (certainly not exhaustive) suggestions for improvement of the English. L18 Nocturnal reactive nitrogen compounds play an important role in regional air pollution*

Changed accordingly.

*24. L27 The concentration of the nitrate radical (NO3) was calculated assuming that. …*

Changed accordingly.

*25. L34 which indicates that reduction of NOx emissions cannot help reduce the nocturnal formation of ONs.*

Changed accordingly.

*26. L42 NO₃ can initiate the removal of many kind of anthropogenic*

Changed accordingly.

*27. L58 the reactions of NO₃ with several BVOCs produce considerable amounts of organic nitrates.*

Changed accordingly.

*28. L207. Nocturnal nitrate radical production rate, P(NO₃), was large, with an average. . .*

Changed accordingly.

*29. L61 The reaction of NO3 with isoprene has a SOA yield of 23.8% (Ng et al., 2008). For the reaction with a monoterpene, such as limonene, the yield can reach 174% at ambient temperatures (Boyd et al., 2017).*

Changed as following; "The reaction of $NO_3$ with isoprene has a SOA mass yield of 23.8% (Ng et al., 2008). For the reaction with monoterpene, such as limonene, the SOA mass yield can reach 174% at ambient temperatures (Boyd et al., 2017)".

*30. L97 the reaction also contributed significantly to NOx.*

Changed accordingly.

*31. L259 by acid displacement*

Changed accordingly.

*32. L260 "however, the photolysis with profound ClNO₂ was still maintained until noon". I think the authors are trying to say that ClNO₂ survived until noon? In this context they should mention the J-values of ClNO₂.*

Yes, we are trying to say the $ClNO_2$ survived until noon. The campaign average J-values of $ClNO_2$ around noon is about $1.7 \times 10^{-4}$ $s^{-1}$. The text changed as following: "However, the $ClNO_2$ can still survive until noon with the averaged daily maximum of $J(ClNO_2)$ to be $1.7 \times 10^{-4}$ $s^{-1}$."

*33. L273-276. This part needs rewriting. I think the gist if this is that the N₂O₅*

*concentration depends on the NO$_2$ level more than on the O$_3$ concentration. If so, please explain why.*

The sentence was rewrote as following: "The N$_2$O$_5$ concentration was highly correlated with NO$_2$ (R$^2$ = 0.81) and the NO$_3$ production rate (R$^2$ = 0.60), suggests the N$_2$O$_5$ concentration was solely response to the NO$_2$ concentration in the background air mass when enough O$_3$ is presented."

*34. L305. Which implies that the ClNO$_2$ formation efficiency.*

Changed accordingly.

*35. L403 The title of this section is misleading. NO3 is not oxidized, but the VOCs. I suggest "NO3-induced nocturnal oxidation of VOCs" or similar.*

Thanks for the suggestion and changed accordingly.

*36. L429 for calculating nocturnal ONs production from NO3 oxidation of isoprene and monoterpene, as well as inorganic nitrate production via N2O5 heterogeneous uptake over the same period.*

Changed accordingly.

---

## Author Response (AR3)

Dear editor,

Thanks for your careful reading and the suggestion of the detailed corrections for improving our manuscript. We corrected accordingly and all revisions were labelled in red in the following differ version.

*Line 273, 308, 331 and 333, replace "transportation" with transport*
Corrected accordingly.

*Line 320, replace "others" with "other factors."*
Corrected accordingly.

*Line 333, replace "contrition" with "contribution"*
Corrected accordingly.

*Line 335, replace "accompanied" with "also observed"*
Corrected accordingly.

*Line 457, replace "the amount quantification uncertainty" with "an uncertainty"*
Corrected accordingly.

*Line 446, 472 and 810, replaced "initialed" with either "initialized" or "derived"*
Corrected accordingly.

[revised manuscript text omitted]